

# A framework for accurate, long-term, global and high resolution observations of tropospheric H$_2$O-$\delta$D pairs — a MUSICA review

M. Schneider[1], A. Wiegele[1], S. Barthlott[1], Y. González[2,3,*], E. Christner[1], C. Dyroff[1,**], O. E. García[2], F. Hase[1], T. Blumenstock[1], E. Sepúlveda[2], G. Mengistu Tsidu[4,5], S. Takele Kenea[4,***], S. Rodríguez[2], and J. Andrey[6,****]

[1]Institute of Meteorology and Climate Research (IMK-ASF), Karlsruhe Institute of Technology, Karlsruhe, Germany
[2]Izaña Atmospheric Research Center, Agencia Estatal de Meteorología (AEMET), Santa Cruz de Tenerife, Spain
[3]Sieltec Canarias, S.L., Hábitat 2, 38204, San Cristóbal de La Laguna, Santa Cruz de Tenerife, Spain
[4]Department of Physics, Addis Ababa University, P.O. Box 1176, Addis Ababa, Ethiopia
[5]Botswana International University of Technology and Science (BIUST) Priv. Bag 16, Palapye, Botswana
[6]Area de Investigación e Instrumentación Atmosférica, INTA, Torrejón de Ardoz, Spain
[*]now at: Department of Earth, Atmospheric, and Planetary Sciences, Massachusetts Institute of Technology, 77 Massachusetts Avenue, Cambridge, MA 02139, USA
[**]now at: Aerodyne Research Inc., 45 Manning Road, Billerica, MA 01821, USA
[***]now at: Department of Physics, Samara University, P.O. BOX 132, Samara, Ethiopia
[****]now at: CNRM-GAME, Météo France and CNRS, Toulouse, France

*Correspondence to:* M. Schneider
(matthias.schneider@kit.edu)

**Abstract.** In the lower/middle troposphere H$_2$O-$\delta$D pairs are good proxies for moisture pathways, however their observation is challenging. The project MUSICA (MUlti-platform remote Sensing of Isotopologues for investigating the Cycle of Atmospheric water) addresses this challenge by integrating remote sensing with in-situ measurement techniques. The aim is to retrieve accurate tropospheric H$_2$O-$\delta$D pairs from the middle infrared spectra measured from ground by the FTIR (Fourier

Transform InfraRed) spectrometers of the NDACC (Network for the Detection of Atmospheric Composition Change) and the thermal nadir spectra measured by IASI (Infrared Atmospheric Sounding Interferometer) aboard the MetOp satellites. In this paper we review the MUSICA framework, present the final MUSICA products, and outline the NDACC/FTIR's and METOP/IASI's potential for observing accurate and consistent H$_2$O-$\delta$D data pairs. First, we briefly resume the particularities of an H$_2$O-$\delta$D pair retrieval. Second, we show that the remote sensing data of the final product version are absolutely calibrated

with respect to H$_2$O and $\delta$D in-situ profile references measured in the subtropics, between 0 and 7 km. Third, we empirically demonstrate that the calibrated remote sensing H$_2$O-$\delta$D pairs can identify different lower/middle tropospheric moisture pathways and advert to the risk of misinterpretations caused by an incorrect processing of such remote sensing data. Fourth, we reveal that the different sensors (NDACC/FTIR instruments, MetOp/IASI-A, and MetOp/IASI-B) provide consistent H$_2$O-$\delta$D pairs for very distinct atmospheric clear sky conditions. Fifth, we document the unique possibilities of the NDACC/FTIR in-

struments for providing long-term records (important for climatological studies) and of the MetOp/IASI sensors for observing diurnal signals on quasi global scale and with high horizontal resolution.



## 1 Introduction

Atmospheric moisture (condensed water and vapour) strongly interacts with solar as well as thermal radiances and distributes energy in form of latent heat. In consequence, it has a wide impact on the atmospheric energy budget and strongly affects circulation on regional and global scale. The insufficient understanding of tropospheric moisture pathways and its
coupling to atmospheric circulation is seen as a major problem for climate system modeling (e.g. Stevens and Bony, 2013; Sherwood et al., 2014) and it is one of the Grand Challenges according to the WCRP (World Climate Research Programme, http://www.wcrp-climate.org/grand-challenges).

In this context, the simultaneous observation of different tropospheric water isotopologues can make a valuable contribution. In contrast to observations of wind, temperature or humidity, which are mainly representative for the location where they are
observed, the ratio between different water isotopologues reflects the physical state of the troposphere upwind of the detected air mass and provides complementary information on moisture uptake, exchange, and atmospheric transportation processes (e.g. Dansgaard, 1964; Gat, 2000; Yoshimura et al., 2004). The isotopologue ratios are typically expressed in the $\delta$-notation, which relates the observed ratio to the standard ratio VSMOW (Vienna Standard Mean Ocean Water). For instance, the $HDO/H_2O$ ratio is typically expressed as $\delta D = \frac{HDO/H_2O}{VSMOW} - 1$. Here and in the following we use $H_2O$ and HDO as equivalent to $H_2^{16}O$ and
$HD^{16}O$, respectively. The consideration of other isotopologues will be specified explicitly (e.g $H_2^{18}O$ or $H_2^{17}O$).

During the last years there has been large progress in observing the tropospheric water vapour isotopologues, whereby remote sensing observations are particularly interesting since they can be performed continuously (for cloud free conditions). Ground-based remote sensing can offer long-term data records. There are the ground-based water vapour isotopologue remote sensing retrievals using the NDACC middle infrared spectra (Schneider et al., 2012, and references therein) and retrievals
that use the TCCON (Total Carbon Column Observing Network) near infrared spectra (e.g. Rokotyan et al., 2014). Tropospheric water vapour isotopologue data sets have also been presented using space-based sensors. There are different research groups using short wave infrared (SWIR) spectra measured by the satellite sensors SCIAMACHY (Frankenberg et al., 2009; Scheepmaker et al., 2015) or GOSAT (Boesch et al., 2013; Frankenberg et al., 2013) as well as the thermal nadir spectra of TES (Worden et al., 2006, 2012) or IASI (Schneider and Hase, 2011; Lacour et al., 2012; Wiegele et al., 2014).
While in the dry upper troposphere and stratosphere $\delta D$ observations alone allow significant conclusions on the moisture pathways from the troposphere to the stratosphere and on stratospheric circulation (e.g. Kuang et al., 2003; Steinwagner et al., 2010), the situation is different in the lower and middle troposphere. There humidity is much more variable and the moisture pathways can be best investigated by analyzing the distribution of the $H_2O$-$\delta D$ pairs (e.g. Galewsky et al., 2005; Noone, 2012; González et al., 2015). Recently, there have been a variety of publications that use remote sensing observations of tropospheric
$H_2O$-$\delta D$ pairs for tropospheric moisture pathway studies: for instance, for estimating the importance of rain recycling (e.g. Worden et al., 2007), for investigating the dynamics of the Madden-Julian oscillation (e.g. Berkelhammer et al., 2012) or for drawing conclusions on vertical mixing processes (e.g. Noone, 2012; Risi et al., 2012a; Sutanto et al., 2015). However, there is only one study so far where attempts have been made for empirically validating these $H_2O$-$\delta D$ pairs (Schneider et al., 2015). Further and more detailed validation efforts for the $H_2O$-$\delta D$ pairs are urgently needed, because an optimal estimation of $H_2O$-



$\delta$D pairs is complex and not the same as an individual optimal estimation of $H_2O$ and HDO amounts (the reason is that the sensitivities for $H_2O$ and HDO are generally different, see Schneider et al., 2006; Worden et al., 2006; Schneider et al., 2012).

A further pending detail with the isotopologue remote sensing data is the unclear bias in $\delta$D, which can significantly compromise their scientific usefulness (e.g. Risi et al., 2012b; Field et al., 2014). For a reliable bias documentation we need vertical isotopologue reference profiles measured by well-calibrated in-situ instrumentation. Actually we are only aware of one campaign (the summer 2013 MUSICA campaign, Dyroff et al., 2015), where such profiles are measured in coincidence with ground- and space-based remote sensing observations and over the wide altitude range where the remote sensors are sensitive.

Removing the shortcomings in tropospheric water vapour isotopologue remote sensing data has been a focus of the project MUSICA (http://www.imk-asf.kit.edu/english/musica.php), which ends in 2016. During the last few years methods for a theoretical characterisation and an empirical validation of the remote sensing $H_2O$-$\delta$D pairs have been developed in the framework of MUSICA. The most important milestones and the corresponding publications are listed in Table 1. In this paper we review the MUSICA activities and summarize the final results of the project. We demonstrate that we are now able to generate accurate, long-term, global and high resolution observational data of tropospheric $H_2O$-$\delta$D pairs. In Section 2 we give a brief review on the theory of the optimal estimation of $H_2O$-$\delta$D pairs and present the retrieval improvements for the final MUSICA data version. Section 3 documents that the final MUSICA data are well-calibrated with respect to in-situ references. Section 4 shows an empirical validation of the $H_2O$-$\delta$D pairs by intercomparing the $H_2O$-$\delta$D signals observed by in-situ instruments, NDACC/FTIR, MetOp/IASI-A and MetOp/IASI-B for different atmospheric situations. Sections 5 and 6 document the unique long-term characteristic of the NDACC/FTIR data and the unique spatial and temporal coverage achievable by MetOp/IASI observations. Section 7 gives a summary.

## 2 Remote sensing of $H_2O$ and $\delta$D

In this section we give a very brief overview on the challenges of tropospheric water vapour isotopologue remote sensing. Then we resume the retrieval approaches as developed in preparation for and continuously improved during the project MUSICA. In addition to the MUSICA products there are other ground- and space-based (non-MUSICA) tropospheric water vapour isotopologue remote sensing products. A brief overview and a short discussion of the differences to the MUSICA products is given in Appendix A.

### 2.1 The challenge

In-situ instruments analyse a clearly defined air mass and from the $H_2O$ and HDO measurements the ratio $\delta$D can be directly calculated. This is more complicated for remote sensing observations. There the amount of $H_2O$ and HDO along the line of sight is retrieved from the measured spectra. The sensitivity of the retrieval depends on the noise in the spectra and on the shape and strength of the different absorption lines. Typically the sensitivity for $H_2O$ is different to the sensitivity for HDO and the retrieved $H_2O$ value represents a different altitude range than the retrieved HDO value ($H_2O$ and HDO have different averaging





kernels). In consequence the $\delta$D values calculated from individual $H_2O$ and HDO retrievals can be rather misleading. Instead of individual retrievals, a combined $H_2O$ and HDO retrieval is needed.

## 2.2  Quasi optimal estimation retrievals of $H_2O$-$\delta$D pairs

A logarithmic scale retrieval of $H_2O$ and HDO together with a constraint of $\ln[\mathrm{HDO}] - \ln[\mathrm{H_2O}]$ has been proposed by
Schneider et al. (2006) and Worden et al. (2006) for generating $\delta$D values by remote sensing techniques. This approach means actually an optimal estimation of $(\ln[\mathrm{H_2O}] + \ln[\mathrm{HDO}])/2$ and $\ln[\mathrm{HDO}] - \ln[\mathrm{H_2O}]$, which are good proxies for $H_2O$ and $\delta$D (Schneider et al., 2012), i.e. it is a quasi optimal estimation of $H_2O$ and $\delta$D. However, it is an individual optimal estimation of $H_2O$ and $\delta$D and thus such product is still not comparable to the $H_2O$-$\delta$D pairs obtained from the in-situ measurements. The reason is that the remote sensing system is more sensitive to atmospheric $H_2O$ than to atmospheric $\delta$D. This problem can be
overcome by an aposteriori processing of the retrieval output so that the $H_2O$ and $\delta$D products are sensitive to almost the same atmospheric altitude ranges. The result of the aposteriori processing is a quasi optimal estimation of $H_2O$-$\delta$D pairs that are comparable to the $H_2O$-$\delta$D pairs obtained from in-situ measurements (once we consider the averaging kernels of the $H_2O$-$\delta$D pairs). The aposteriori processing and its motivation are explained in detail in Sect. 4 of Schneider et al. (2012).

## 2.3  Remote sensing within MUSICA

MUSICA offers two types of water vapour remote sensing products. Type 1 is the output of an individual optimal estimation of $H_2O$ and $\delta$D and offers optimal $H_2O$ data, but limited possibilities for isotopologue studies. Type 2 is the aposteriori processed product, i.e. it is a quasi optimal estimation of $H_2O$-$\delta$D pairs (meaning that $H_2O$ and $\delta$D have almost the same averaging kernels). The separation in Type 1 and 2 is explained in detail in Sect. 4 of Schneider et al. (2012). If not stated otherwise we refer here always to Type 2, which is the product that is most useful for isotopologue studies.

### 2.3.1  The ground-based products

The MUSICA ground-based remote sensing retrieval uses the middle-infrared spectra recorded within the NDACC (information on the NDACC infrared working group can be found at: www.acom.ucar.edu/irwg/). The NDACC spectra are of very high spectral resolution ($0.005\,\mathrm{cm}^{-1}$), offer $H_2^{16}O$, $H_2^{18}O$ and $HD^{16}O$ absorption lines of similar strengths. For the previous retrieval version we used 11 spectral windows with lines of water vapour isotopologues (see Fig. 2 of Schneider et al., 2012) and three
windows with $CO_2$ lines (beneficial for the atmospheric temperature retrievals).

For the final MUSICA retrieval version (v2015) we removed the windows with strong $H_2^{16}O$ lines and replaced it by windows with weaker lines. By this modification we want to make sure that even for very humid sites the observed spectral lines do not saturate. Furthermore, we add a second window where a $H_2^{18}O$ signature dominates. The nine spectral water vapour isotopologue windows are depicted in Fig. 1 for a typical observation of the FTIR spectrometer at Izaña. In addition we fit
the three spectral windows with the $CO_2$ lines (the same as for the previous retrieval version: between 2610.35-2610.8 $\mathrm{cm}^{-1}$, 2613.7-2615.4 $\mathrm{cm}^{-1}$ and 2626.3,2627.0 $\mathrm{cm}^{-1}$; not shown). All these spectral windows are covered by the NDACC filter #3.





For v2015 we perform an optimal estimation of $(\ln[H_2^{16}O] + \ln[H_2^{18}O] + \ln[HD^{16}O])/3$, $\ln[H_2^{18}O] - \ln[H_2^{16}O]$, and $\ln[HD^{16}O] - \ln[H_2^{16}O]$. These are good proxies for $H_2O$, $\delta^{18}$ and $\delta D$. This is a further development of the previous retrieval version (there the optimal estimation was made for $(\ln[H_2^{16}O] + \ln[HD^{16}O])/2$ and $\ln[HD^{16}O] - \ln[H_2^{16}O]$). The further development means a cross constraint between the three water vapour isotopologues $\ln[H_2^{16}O]$, $\ln[H_2^{18}O]$, and $\ln[HD^{16}O]$. For this purpose the transformation operator $\mathbf{P}$, that has been extended with respect to the previous versions. For v2015 the operator $\mathbf{P}$ is as follows:

$$\mathbf{P} = \begin{pmatrix} \frac{1}{3}\mathbb{I} & \frac{1}{3}\mathbb{I} & \frac{1}{3}\mathbb{I} \\ -\mathbb{I} & \mathbb{I} & 0 \\ -\mathbb{I} & 0 & \mathbb{I} \end{pmatrix} \qquad (1)$$

Here $\mathbb{I}$ stands for an identity matrix. Details on the importance of $\mathbf{P}$ for the data characterisation and the aposteriori treatment are explained in Sects. 3 and 4 of Schneider et al. (2012).

In addition to the cross-constrained fit of the aforementioned three water vapor isotopologues we perform simultaneous but individual fits (no cross-constraints) for profiles of the water vapour isotopologue $H_2^{17}O$, the temperature and the interfering species $CO_2$, $O_3$, $N_2O$, $CH_4$ and $HCl$.

For the previous retrieval version we used the HITRAN 2008 spectroscopic water vapour line parameters Rothman et al. (2009) and adopted it for speed-dependent Voigt line shapes (Schneider et al., 2011). For the ground-based FTIR retrieval this consideration of a non-Voigt line shape parameterisation becomes important, because of the very high resolution spectra (Full Width Half Maximum of the instrumental line shape of about $0.005\,\mathrm{cm}^{-1}$).

For v2015 we work with the HITRAN 2012 parameters (Rothman et al., 2013) and optimised it for speed-dependent Voigt line parameterisation. For the further optimisation of the HITRAN parameters we use the MUSICA in-situ profile references, the coincident FTIR spectra, and FTIR spectra measured at three rather distinct sites (Izaña on Tenerife Island, subtropical ocean, Karlsruhe, central Europe, and Kiruna, northern Sweden). This method for line parameter optimisation by means of atmospheric spectra is described in detail in Schneider and Hase (2009) and Schneider et al. (2011). We changed line intensities and broadening parameters by about 5-10%, which is in agreement with the uncertainty values as given in the HITRAN parameter files. More details on the modification of the parameters of the lines as shown in Fig. 1 are given in the Appendix B.

The Type 1 MUSICA NDACC/FTIR product consists of water vapour profiles for the lower, middle, and upper troposphere (DOFS of almost 3). The Type 2 product offers consistent $H_2O$-$\delta D$ pairs, which are sensitive to the lower and the middle troposphere, whereby it is possible to reasonably separate both altitude regions (degree of freedom for signal, DOFS, is typically about 1.8). Error estimations are discussed in detail in Schneider et al. (2012). The random error is about 2% for $H_2O$ and 25‰ for $\delta D$. Already for small uncertainties of 1% and 2% for line intensity and pressure broadening parameters the systematic errors in $H_2O$ and $\delta D$ "profiles" can reach 10% an 150‰, respectively. An empirical study indicates that for the v2015 data the bias is actually much smaller (see Sect. 3).

### 2.3.2 The space-based products

The basic MUSICA MetOp/IASI retrieval setup is presented and analysed in detail in Schneider and Hase (2011) and Wiegele et al. (2014). It has been developed in consistency to the NDACC/FTIR retrieval. It uses a broad spectral window (1190-1400 $\mathrm{cm}^{-1}$),





whereby the humidity and $\delta$D proxies $((\ln[H_2O] + \ln[HDO])/2$ and $\ln[HDO] - \ln[H_2O])$, as well as temperature, and the interfering species $CH_4$, $N_2O$, $CO_2$, and $HNO_3$ are simultaneously fitted.

The previous retrieval version works with HITRAN 2008 spectroscopic water vapour line parameters (Rothman et al., 2009). For v2015 we use the HITRAN 2012 parameters (Rothman et al., 2013) and modified the line intensities ($S$) for the HDO ab-

sorption signatures by $+10\%$. We also tested modification of $S$ for the $H_2O$ signatures and changes of $\gamma_{air}$ (pressure broadening parameter), but finally found that a modification of $S$ of HDO works most effectively for correcting biases in $\delta$D. In this context, we would like to remark that the bias correction as suggested for TES (Worden et al., 2011; Herman et al., 2014) is also consistent with a positive change of $S$ of HDO.

The retrieval provides a Type 1 product of $H_2O$ profiles being sensitive to variations between the surface and about $15\,km$

altitude (DOFS of about 4). The Type 2 product (consistent $H_2O$-$\delta$D pairs) has a DOFS of typically 0.5-1.0, whereby the sensitivity is mainly limited to the middle troposphere. The random error is about 5% for $H_2O$ and 20‰ for $\delta$D and the systematic error can easily reach 5% and 50‰ for $H_2O$ and $\delta$D, respectively (in case of a 5% error in the spectroscopic line intensity parameters). However, an empirical study as shown in Sect. 3 shows that for v2015 the bias is actually much smaller.

### 2.3.3   Uniform apriori settings

For v2015 we work with the same globally constant water vapour isotopologue apriori data for all ground-based retrievals (different globally distributed NDACC/FTIR stations) and for all MetOp/IASI retrievals (for the whole globe). Thereby we assure that observations at different locations are not affected by the use of different apriori and therefore an interpretation of regional differences (e.g. latitudinal gradients) becomes rather straight-forward. This is a further development of the previous retrieval versions, where we used different apriori data for the (sub-)tropics, the mid-latitudes, and the polar regions.

The used apriori profiles are mean values of LMDZiso calculations (Risi et al., 2012b) and are as depicted in Fig. 2 of Lacour et al. (2012). As $H_2O$ apriori variability we assume 75% in the boundary layer, 150% in the middle and upper troposphere, and 30% in the stratosphere. For $\delta$D the respective variability values are 60‰, 120‰, and 50‰. The apriori covariances are then calculated by assuming a correlation length of $2\,km$ in the boundary layer, $4\,km$ in the middle and upper troposphere, and $8\,km$ in the stratosphere.

## 3   Accurate remote sensing products

The summer 2013 MUSICA campaign generated unique reliable $H_2O$ and HDO (and $\delta$D) in-situ reference profiles on six days between sea surface and 6-7 km altitude (Dyroff et al., 2015; Schneider et al., 2015). From these $\delta$D profiles we calculate $\delta^{18}$ and $H_2^{18}O$ profiles by assuming $\delta D = 8 \times \delta^{18}$ (a rough estimations of $\delta^{18}$ or $H_2^{18}O$ is needed for validating the v2015 NDACC/FTIR product). The in-situ profiles have been measured in coincidence to high resolution ground-based FTIR obser-

vations and to IASI observations and are unique for documenting the bias in the remote sensing data, because they cover the whole altitude range from the surface up to 6-7 km. Appendix C provides a brief discussion on the importance of reaching high ceiling altitudes.





In this section we show that the MUSICA v2015 remote sensing products are well calibrated with respect to the reference data. We would like to remind that there is no similar study where the accuracy of simultaneous tropospheric $H_2O$ and $\delta D$ remote sensing observations is documented by such direct comparison to coincident reference profiles.

## 3.1 NDACC/FTIR

In a first study with data from the previous MUSICA NDACC/FTIR retrieval version, we found a bias of 25-70‰ for $\delta D$ with respect to the profile references (see right panels in Figs. 9 and 10 of Schneider et al., 2015). That exemplary study has been made with a limited data set (only one exemplary remote sensing observation per day). For the v2015 product we perform a comprehensive empirical bias assessment. For this purpose we compare the ground-based FTIR data obtained for all optimal coincidences (about ten observations each day between 8:15 and 9:45 UT) with the reference data (one profile per day, YYMMDD: 130721, 130722, 130724, 130725, 130730, and 130731).

Figure 2 shows the plots for the correlations between the reference and the FTIR data. The upper panel depicts the comparison for the lower free troposphere and the bottom panel for the middle free troposphere. The references are constructed from the in-situ profile measurements (surface up to ceiling altitude of 6-7 km) and a climatology for higher altitudes by convolution with the FTIR averaging kernels (for the climatology at higher altitudes we use the apriori data as discussed in Sect. 2.3.3). The technical details for this comparison are the same as for the exemplary study of Schneider et al. (2015). The large error bars on the reference data are mainly due to the unknown humidity and $\delta D$ values above the aircraft's ceiling altitude (for a more detailed discussion see Appendix C).

We observe a reasonable correlation and no significant bias between the reference data and the MUSICA NDACC/FTIR v2015 data. We also see variations in the FTIR data measured between 8:15 and 9:45 UT on a single day, which cannot be attributed to changes in the averaging kernels (there is no similar variation seen in the smoothed reference data). This variation is seen in the lower free troposphere and in the middle free troposphere and is very likely a true variation in the free tropospheric humidity and $\delta D$ fields. The relatively high variability in the atmospheric state is a problem when comparing the different measurements, because we do not well know whether the different measurements detect air masses with the same atmospheric characteristic. For a detailed discussion on coincidences between the aircraft-based in-situ and the remote sensing measurements please refer to Appendix C.

The Tables 2 and 3 resume the results of the bias assessments. The calculated mean NDACC/FTIR biases are not significant if we take into account the confidence range of these assessments. The confidence range is calculated as the standard deviation of the bias divided by $\sqrt{N_R - 1}$ (with $N_R$ being the number of independent reference observations). For the lower troposphere we are able to determine the bias with a confidence of 12.4% and 16.6‰ for $H_2O$ and $\delta D$, respectively. In the middle troposphere the assessment is even more reliable. There the confidence ranges are 8.2% and 7.4‰ for $H_2O$ and $\delta D$, respectively. In summary, the lower tropospheric bias is very likely somewhere between $-10\%$ and $+15\%$ for $H_2O$ and between $-30$‰ and $+5$‰ for $\delta D$. The middle tropospheric bias is very likely between $-10\%$ and $+8\%$ for $H_2O$ and between $-10$‰ and $+5$ for $\delta D$.





## 3.2 MetOp/IASI

In an exemplary study with the previous MUSICA MetOp/IASI retrieval version Schneider et al. (2015) reported a bias between in the IASI $\delta$D product and the $\delta$D reference of about 60‰ (see Fig. 11 therein). For the comprehensive bias assessment of the v2015 data we follow procedure as described in the context of this exemplary study.

Figure 3 shows the correlation plots between the reference and the MetOp/IASI v2015 data. The data points plotted as smaller symbols and with black error bars are for non-ideal coincidences and the rest of the data points are for good coincidences (Appendix C gives more details on the coincidences and the error bars). For this comparison we smooth the reference profile data (in-situ measurements and climatological data above ceiling altitude) with the respective IASI averaging kernels, i.e. the small variations of the reference data on an individual day are due to the varying averaging kernels (for instance, the small

variability of the green dots for day 130724 in parallel to the x-axis). The respective variations of the IASI data (variations in parallel to the y-axis) are due to variations in the sensitivity of IASI (variation in the kernels) and due to variations in the real atmospheric state encountered at the different observational pixels. The in-situ data and the remote sensing data observed for good coincidences are well correlated.

We are able to assess the bias for the middle tropospheric IASI data with a confidence of 3.7% and 7.9‰ for $H_2O$ and $\delta$D,

respectively (see Table 3). The obtained mean bias values lie within these confidence ranges (for $H_2O$) or are only very slightly outside this range (for $\delta$D), meaning that the actual biases are very likely between $-2.5\%$ and $+5\%$ for $H_2O$ and between 0‰ and $+15‰$ for $\delta$D.

## 4    Validation of $H_2O$-$\delta$D pairs

An individual validation of $H_2O$ and $\delta$D (as shown in the previous section) is important for documenting that the data are

bias free. However, it is not sufficient. It is the tropospheric $H_2O$-$\delta$D distribution that gives insight into tropospheric moisture pathways and it is the distribution of these pairs that has to be validated. Wiegele et al. (2014) and Schneider et al. (2015) presented approaches for such kind of validation exercises. Here we present further refined $H_2O$-$\delta$D pair validation exercise and compare in-situ, MUSICA v2015 NDACC/FTIR, MUSICA v2015 MetOp/IASI-A, and MUSICA v2015 MetOp/IASI-B data sets.

### 4.1    Moisture pathways to the North Atlantic subtropical free troposphere

Table 4 lists the three moisture pathways to the free troposphere that are prevailing in the surroundings of Tenerife. They have distinct isotopologue fingerprints (González et al., 2015) and can be used for validating the middle tropospheric $H_2O$-$\delta$D pairs.

Generally the free troposphere in the subtropics receives air that has been transported from higher latitudes and altitudes and subsides to the subtropics (e.g. Galewsky et al., 2005). In the following we call this pathway "ATL, desc". However, the

summertime free troposphere close to West Africa is often affected by the Saharan Air Layer (SAL). The SAL is a well-mixed planetary boundary layer that can expand up to 6-7 km and has its origin in the strong vertical mixing (dry convection) over





the summertime Sahara. This dry convection process mixes boundary layer air with free tropospheric air. The SAL is then often advected westward over the Atlantic, where it can be identified by high dust concentrations (Rodríguez et al., 2011) and increased humidity levels. The free troposphere above Tenerife is also particularly humid when the air has been transported from lower altitudes over the tropical/subtropical Atlantic (González et al., 2015). This mainly occurs in the late summer and

early autumn. We call this pathway in the following "ATL, asc".

Figure 4 depicts $H_2O$-$\delta D$ contour plots made from different data sets. Such contour plots allow a more detailed characterisation of distribution of the $H_2O$-$\delta D$ pairs than the simple data point plots as shown in Schneider et al. (2015). We present these plots on a logarithmic scale for $H_2O$ and maintain the scale for $\delta D$ (the $\delta D$ scale is in a first a approximation the same as a $\ln[HDO] - \ln[H_2O]$ scale). These are the scales on which the optimal estimation of the remote sensing products is performed.

This largely facilitates the interpretation of the remote sensing data, because then the Type 2 kernels are very similar on the x- and y-scales. Furthermore, on these scales a Rayleigh process will become visible by an almost linear distribution.

The blue and green contours corresponds to observations of Atlantic air masses. Blue is for moisture arriving from higher altitudes and latitudes and green for moisture that has been transported from lower latitudes and altitudes. The red contours are for observations that correspond to SAL conditions (air masses that experienced dry convection over the African continent).

The different pathways can be identified by using aerosol measurements and trajectory calculations (a detailed discussion on the methods that allow identifying the different pathways is given in González et al., 2015).

The top panel shows the in-situ reference data as measured during night (midnight until one hour after sunrise) at two different mountain stations on Tenerife Island between March 2012 and May 2015. We observe that the $H_2O$-$\delta D$ distribution is very different for the three prevailing moisture pathways.

For air descending from the Northern Atlantic (blue contours) the data points are well distributed between a typical Rayleigh line (gradual dehydration due to condensation, cyan line) and mixing lines (mixing of two end members with $H_2O$-$\delta D$ according to the Rayleigh line, black lines). These water masses have gone through different condensation and mixing processes.

For air ascending from the tropical/subtropical Atlantic (green contours) the air is more humid and the $H_2O$-$\delta D$ pairs group generally around the Rayleigh line with some data points lying significantly below the Rayleigh line. These water masses are

strongly depleted in HDO, which indicates rain re-evaporation or gradual dehydration (Rayleigh distillation) after evaporation over a warm Ocean.

For SAL conditions (red contours) the air is also humid but HDO is significantly enriched if compared to the typical Rayleigh distribution. This can be well explained by the mixing of planetary boundary layer humidity with middle/upper tropospheric humidity.

The middle panels show the $H_2O$-$\delta D$ contours as obtained from the remote sensing data sets (Type 2, i.e. consistent $H_2O$-$\delta D$ pairs). The left panel shows the Izaña FTIR data as observed between 2005 and 2014 (about 7400 individual FTIR observations) and the right panel the IASI data obtained from IASI-A and -B spectra measured in a $2° \times 2°$ area around Tenerife between 2007 and 2013 (about 10000 individual IASI measurements).

We observe that the $H_2O$-$\delta D$ distributions for the three different pathways show strong similarity to the distributions obtained

from the in-situ data, which confirms that the v2015 MUSICA remote sensing data are reasonably-well bias corrected and




proves that they are capable of tracking different moisture pathways. The smaller variability in the remote sensing data is due to the fact that they represent averages for layers of several kilometers (please refer to the averaging kernel plots as shown in Figs. 3 and 4 of Schneider et al., 2015). For the MetOp/IASI data the variability is in particularly small for dry air (blue contours), because the drier the atmosphere the lower IASI's sensitivity for middle tropospheric $H_2O$-$\delta D$ pairs.

The bottom panels show the $H_2O$-$\delta D$ contours as obtained from the remote sensing Type 1 products, i.e. individual optimal estimation of $H_2O$ and $\delta D$. For the Type 1 product $H_2O$ and $\delta D$ are not sensitive for the same air mass and the retrieval response is much more sensitive to atmospheric $H_2O$ than to $\delta D$ variations. This is clearly seen in the MetOp/IASI contour plots. The large variability in $H_2O$ together with the reduced variability in $\delta D$ (limited sensitivity) leads to artificially low slopes in the $H_2O$-$\delta D$ plots and using the Type 1 product can easily lead to misleading conclusions.

Concretely the IASI Type 1 data for air descending from the North Atlantic (blue contour lines) describe a line in parallel to the $H_2O$-axis (variation of $H_2O$, but almost no variation of $\delta D$), indicating drying of a humid air mass by mixing with very dry air. This is in clear contrast to the in-situ data, which suggests drying by condensation and only partly by mixing with dry air.

     The IASI Type 1 distributions for SAL conditions (red contours) is relatively close to the Rayleigh line, which is again in disagreement to the in-situ reference (there and in the corresponding IASI Type 2 distribution the data points group around a

mixing line).

     Similar observations are made for the NDACC/FTIR Type 1 product. There the retrieval is also more sensitive to $H_2O$ than to $\delta D$ variations, but the difference is not that substantial as for the MetOp/IASI Type 1 product.

## 4.2    $H_2O$-$\delta D$ extremes on global scale

     Figure 4 shows validations of $H_2O$-$\delta D$ pairs for a subtropical site. In order to perform a similar study for other sites we would

need respective middle tropospheric in-situ references, which are not available. The ISOWAT profile and surface-based Izaña and Teide Picarro in-situ references as observed in the surroundings of Tenerife are unique and a generation of similar data sets for middle or high latitudes would be expensive (it would require a large number of aircraft campaigns).

     Here we show a comparison between NDACC/FTIR, MetOp/IASI-A, and MetOp/IASI-B, which is of global validity. Our argument is that a global agreement between the different remote sensing data sets would suggest that the in-situ validations

made for Tenerife Island are of global validity.

     For this kind of validation we work with $H_2O$-$\delta D$ extremes. For this purpose we identify anomalous or extreme $H_2O$-$\delta D$ distributions in one remote sensing data set and document to what extent these extremes are seen in another remote sensing data set (by comparing coincident observations). The validation approach with $H_2O$-$\delta D$ extremes has been first presented by Wiegele et al. (2014), which should be consulted for more details.

### 4.2.1    NDACC/FTIR versus MetOp/IASI

     We compare the FTIR and IASI data for three rather different sites: Tenerife (subtropical Atlantic), Karlsruhe (central Europe) and Kiruna (Northern Scandinavia). At these sites we have ground-based FTIR observations of NDACC that contribute to MUSICA and we performed IASI retrievals in an area of $200\,\mathrm{km}$ around the FTIR locations. Figure 5 shows the $H_2O$-$\delta D$





distributions as retrieved at the three sites from coincident FTIR and IASI measurements. As temporal coincidence criterium we required that the two measurements were made within 1 hour. The left column of plots shows data for Tenerife (coincidences between 2007 and 2013), the central column data for Karlsruhe (coincidences between 2010 and 2013), and the right column data for Kiruna (coincidence between 2007 and 2012).

The first row of plots depicts the FTIR data, the second row of plots shows the IASI data, and the third row of plots the FTIR data smoothed by the IASI averaging kernels. In all plots we show retrievals for 5 km altitude.

The grey data points represent all data. The FTIR observations that show unusual low or strong HDO depletion (high or low $\delta$D values) are marked by red and green colour, respectively. These "anomalies" or "extremes" have been identified by a second order least squares fit to the $\ln[H_2O]$-$\delta$D distribution. The 10% of the data points that have the largest positive/negative

$\delta$D difference to the regression curve are defined as the extreme values.

First, comparing the $H_2O$-$\delta$D distribution relative to the unique apriori point, we see a good agreement between both data sets. In both data sets and from Tenerife via Karlsruhe to Kiruna the water masses get generally more and more depleted in HDO and the $\ln[H_2O]$-$\delta$D slopes become more and more shallow. Second, both data sets reveal very similar anomalies. If the FTIR observes an anomalously weak depletion, IASI also does (red dots in the IASI plots are situated at the upper end of the

$\delta$D distribution). The same is true for the anomalies with strong depletion (green dots).

### 4.2.2  IASI-A versus IASI-B

Since 2013 two IASI instruments (A and B) on two different satellites (MetOp-A and -B) provide operational spectra. Their respective overpasses take place typically within 30 minutes, which offers very good conditions for cross-validating the IASI-A and -B products.

Figure 6 depicts $H_2O$-$\delta$D distributions considering all valid observations on 16. August 2014 (left columns for the morning overpass and right columns for the evening overpass). The colours are as in Fig. 5. The grey data points show all data, the red data points mark the observations that have been identified in the IASI-A data as a positive $\delta$D extreme and the green data points mark the observations that correspond to a negative IASI-A $\delta$D extreme.

The top panels show the IASI-A data. These data are used for identifying the extremes and the red and green data points are

of course separated.

The bottom panel shows the IASI-B data and the green and red colour marks the IASI-B observations that are made in coincidence to the extreme IASI-A observations. The coincidence criteria were measurements within one hour and within an area of $0.25° \times 0.25°$. We find that IASI-B detects very similar $\delta$D extremes as IASI-A, which demonstrates the good global consistency of the IASI-A and IASI-B $H_2O$-$\delta$D pairs. In summary, we can use the IASI-A and -B products as a uniform and

consistent data set.





## 5 Consistent long-term observation with NDACC/FTIR

Ground-based FTIR high resolution solar absorption spectra have been measured within the NDACC since many years and can be used for generating long-term data sets of tropospheric $H_2O$-$\delta D$ pairs (Schneider et al., 2012). For MUSICA the principal investigators of the individual FTIR stations send the spectra to the MUSICA retrieval team where they are centrally evaluated.

This strategy assures highest consistency between the retrieval products for the different stations.

All the data pass through a quality filter with different criteria. First, we require a DOFS for the three water vapour isotopologues ($H_2^{16}O$, $HD^{16}O$ and $H_2^{18}O$) together of at least 4.0. Second, we analyse the position of solar lines with respect to terrestrial lines and require a $\Delta\nu/\nu$ within $5 \times 10^{-6}$ ($\Delta\nu$ is the difference in the line shift of solar and terrestrial lines and $\nu$ the position of the solar line, both given in cm$^{-1}$). This method allows for excluding observations made with incorrect pointing

of the solar tracker (Gisi et al., 2011). Third, we perform $XCO_2$ retrievals using the same spectra as for the water vapour isotopologue retrievals. We compare the retrieved $XCO_2$ values with $XCO_2$ as obtained from a multi parameter model for $XCO_2$ (Barthlott et al., 2015). We require that the measured and modelled $XCO_2$ data agree within 2%.

The number of stations contributing to the MUSICA activities is gradually increasing and the data sets have been updated. Table 5 gives an overview of the NDACC/FTIR sites that now contribute to the MUSICA activities. The stations are well

distributed from the Arctic to the Antarctic and in some occasions offer data since the late 1990s. A further extension of this data set to other sites or for some stations to measurements made in the beginning of the 1990 is feasible but has not been possible with the funds available for the MUSICA project.

The NDACC/FTIR activities complement the surface-based in-situ isotopologue monitoring activities. While the data obtained from the latter represent near surface small-scale variations which are often difficult to be captured by models, the

MUSICA NDACC/FTIR isotopologue data are representative for different altitudes and for larger scale processes (the data represent vertical layers averaged over 2-5 km, see typical averaging kernels in Fig. 3 of Schneider et al., 2015). Due to their long-term data characteristics the NDACC/FTIR data are in particularly interesting for climatological studies.

Figure 7 gives an example of the seasonal cycles in the $H_2O$-$\delta D$ distributions (around 5 km altitude) obtained from the observations made on Tenerife/subtropical North Atlantic (15 years: 1999-2014) and in Addis Ababa/East Africa (4 years:

2009-2013). Blue is for October to December (OND), green for January to March (JFM), purple for April to June (AMJ) and red for July to September (JAS). The main intention of this Figure is to briefly demonstrate the potential of the NDACC/FTIR data (a scientifically more detailed discussion is out of the scope of this paper).

The upper panels show Type 2 data. For the subtropical North Atlantic (left panel) the seasonal cycle can be well understood by the seasonal distribution of the different pathways as discussed in Sect. 4 in the context of Fig. 4: in summer (JAS) the vapour

is most enriched in HDO. This is due to the SAL events that are then most frequent. After summer/autumn (OND) the free tropospheric vapour concentrations are occasionally as high as during summer, but significantly more depleted in HDO, which can be understood by moisture transport from the lower troposphere of lower latitudes. In winter (JFM) the free troposphere is driest, whereby the $H_2O$-$\delta D$ data points are clearly above the depicted exemplary Rayleigh line, indicating prevailing transport from high latitudes and altitudes.





Over East Africa (right panel) the air is generally more humid and less depleted in HDO than over the subtropical North Atlantic. We clearly observe different $H_2O$-$\delta D$ distributions for the different seasons. During JAS there is rain season in Addis Ababa and FTIR measurements are not possible. After the rain season (OND, blue) the vapour is most depleted in HDO. Then evaporation of precipitated surface water might be more important as free tropospheric moisture source than during the other

seasons. Between January and March (JAM, green) the vapour concentrations remain similar as during the previous months, but the air becomes more enriched in HDO, which might indicate increased importance of mixing with humid boundary layer air and reduced importance of condensation processes. For April to June (AMJ, purple) air gets more humid, but $\delta D$ remains almost constant, indicating to further moistening due to even stronger mixing with humid boundary layer air.

The lower panel of Fig. 7 shows the Type 1 data. There $H_2O$-$\delta D$ slopes are reduced if compared to the respective Type 2

slopes. This can lead to misleading conclusions (see also the discussion in the context of Fig. 4).

## 6  Quasi global and high resolution observations with MetOp/IASI

IASI sensors are aboard the MetOp satellites, which is a series of three satellites (MetOp-A, -B and -C) for covering the time period from 2006 to the beginning of the 2020s. MetOp has 14 orbits per day at about 817 km altitude, which, together with the swath width of about 2200 km of the IASI instruments, leads to a quasi global coverage of morning overpasses (at about

10:00 local time) as well as evening overpasses (at about 22:00 local time). The IASI ground pixel at nadir has a diameter of only 12 km. MetOp-A with IASI-A was launched in October 2006 and MetOp-B with IASI-B in September 2012. Currently, both IASI instruments are operative.

### 6.1  Spatial and temporal resolution and coverage

For one morning or evening overpass the IASI-B swaths typically complement the area left out by the IASI-A swaths, and

vice versa. Since the MUSICA IASI-A and -B data are very consistent (see Fig. 6) we can treat them as a uniform data set and create extremely dense global data point maps for each daily morning and evening overpass. Figures 8 and 9 depict typical maps for single day morning overpasses during boreal winter and summer, respectively (similar maps can be created for evening overpasses, plots not shown). Please recall that one data point represents a ground pixel of 12 km diameter (at nadir). The areas with missing data are cloudy areas or correspond to scenes where the retrieval has rather low sensitivity.

### 6.2  Sensitivity filter

In Figs. 8 and 9 we only plot data points for which the 5 km altitude (about 500 hPa) retrieval output is sensitive to tropospheric variations that take place within a relatively deep layer (extension of 5 km). The height region around 5 km altitude is generally most sensitive with respect to the $H_2O$-$\delta D$ pairs. However, occasionally (e.g. for an extremely dry or humid troposphere) the sensitivity peaks at other heights.

To filter out such data we set up a matrix in representation of the atmospheric covariances (the matrix's elements represent the different altitude levels). This matrix ($\mathbf{S_c}$) has unity values on the diagonal and the outer-diagonal elements are obtained



by assuming an inter-level correlations length of 5 km. Then we calculate the error covariance in the retrieved data as follows: $\mathbf{S_c'} = (\mathbf{A} - \mathbf{I})\mathbf{S_c}(\mathbf{A} - \mathbf{I})^T$. Here, $\mathbf{A}$ is the averaging kernel matrix and $\mathbf{I}$ the identity matrix. We filter out retrievals, for which the 5 km diagonal element of $\mathbf{S_c'}$ is larger than $0.5^2$, i.e. we require that at least 50% of the middle tropospheric variation is seen in the value as retrieved at 5 km.

The cloud and 5 km sensitivity filter leaves us with about 140000 valid data points for each single morning and evening overpass (IASI-A and -B together). And each of these data points represents the middle tropospheric situation of a small area (12 km diameter at nadir).

## 6.3 Diurnal cycle signals

Morning overpasses of the IASI instruments are at about 10 local time and evening overpasses at about 22 local time. This
offers unique possibilities for a space-based observation of diurnal time scale signals in the tropospheric water cycle. Here, we show an example for the Sahara desert (22.5 to 32.5°N and 10°W to 30°E).

The SAL events as observed over the Atlantic ocean are discussed in the context of Fig. 4. Actually, the dry convection process that is responsible for the distinct $H_2O$-$\delta D$ distribution under SAL conditions takes place over the Sahara desert. The strong heating of the Earth's surface during day in summer is the main driver of these processes, which should be manifested
by a pronounced diurnal cycle over the Sahara. In Fig. 10 on the top panels we depict the $H_2O$-$\delta D$ distribution for three consecutive boreal winter and summer days (left and right panels, respectively). For the winter observations no difference between morning and evening can be identified. This is different for the summer observations, where the morning and evening $H_2O$-$\delta D$ distribution differ significantly. For the morning overpasses we observe a fractionation that is similar to the situation in winter ($\delta D$ values between -300‰ and -200‰). For the evening overpass the $\delta D$ values veer further away from the Rayleigh
line and group around a line that simulates mixing between planetary boundary layer air and middle free tropospheric air ($\delta D$ values between -250‰ and -140‰). This evening distribution is very similar to the distribution that IASI typically observes over the Atlantic under SAL conditions (red contours in the right graph of the second row of panels of Fig. 4).

The observation of such diurnal time scale processes of the water cycle is possible due to IASI's unique high spatial and temporal coverage. This coverage is not achieved by any other space-based instrument that can detect tropospheric water
vapour isotopologues.

The bottom panels of Fig. 10 show the same as the top panels, but for the Type 1 product instead of the Type 2 product. We see that the distribution of the $H_2O$-$\delta D$ pairs changes significantly. For Type 1 and for boreal winter there is a significant number of data points above the mixing lines (air rather enriched in HDO, which artificially points to mixing of very dry and very humid air masses) and for boreal summer there is a significant number of data points below the Rayleigh line (air rather
depleted in HDO, which artificially indicates to rain re-evaporation or evaporation over a very warm ocean). This comparison between the Type 1 and Type 2 results again demonstrate the difficulty of correctly interpreting the Type 1 product, because there $H_2O$ and $\delta D$ represent different air masses. We strongly recommend the use of Type 2 products where $H_2O$ and $\delta D$ represent the same air mass (optimal estimated $H_2O$-$\delta D$ pairs).



## 7   Summary

Water vapour isotopologue remote sensing products are complex data products. In this paper we review the progress achieved within the MUSICA project. The developments made within the MUSICA framework allow for generating remote sensing $H_2O$-$\delta D$ pairs for the troposphere with a high and well-understood data quality and consistent to in-situ references.

5      We present the final version (v2015) of the MUSICA ground- and space-based remote sensing product (generated from NDACC/FTIR and MetOp/IASI spectra, respectively). Compared to previous versions, v2015 improves the consistency between the different locations (uniform apriori for all retrievals and improved spectral windows for the NDACC/FTIR retrievals). Furthermore, for v2015 we calibrate the spectroscopic parameters in agreement with the uncertainty ranges as given in the HITRAN database. This calibration significantly reduces the $\delta D$ bias in the remote sensing products. While in the previous MUSICA versions we found strong indications of a bias between $+25‰$ and $+70‰$, the bias is almost negligible for the v2015 data. This comprehensive bias assessment has been possible by using the unique $H_2O$ and $\delta D$ profile references generated during the summer 2013 MUSICA campaign between the surface and 7 km altitude in the surroundings of Tenerife Island and in coincidence with the remote sensing observations.

Tropospheric $\delta D$ values are most interesting for science if provided together with $H_2O$, i.e. in form of $H_2O$-$\delta D$ pairs. MUSICA's surface-based in-situ measurements made on Tenerife Island at 2400 and 3550 m a.s.l. (Subtropical North Atlantic) provide a continuous free tropospheric in-situ reference record of $H_2O$-$\delta D$ pairs. We use this record for validating the remote sensing data. We find that the in-situ $H_2O$-$\delta D$ pairs and the optimally estimated and calibrated remote sensing $H_2O$-$\delta D$ pairs are similarly distributed and consistently capture the three principle moisture pathways to the subtropical free troposphere: transport from the upper troposphere of the extra-tropics, transport from the lower troposphere over the subtropical/tropical Ocean, and uplift via dry convection over the Sahara followed by advection over the Atlantic. We stress the importance of using the optimally estimated $H_2O$-$\delta D$ pairs (we call it Type 2 product, which is obtained from an aposteriori processing) instead of using data obtained by an individual optimal estimation of $H_2O$ and $\delta D$ (Type 1 product). Usage of the latter manifests in artificially too shallow slopes in the $H_2O$-$\delta D$ plots, which can easily result in misleading interpretations of the $H_2O$-$\delta D$ plots. This is especially the case for the IASI product due to the limited sensitivity with respect to $\delta D$.

25      We show that the space- and ground-based MUSICA v2015 data are rather consistent on global scale. First, there is no significant bias between both data sets and second, the space- and ground-based products consistently detect extremes in the $H_2O$-$\delta D$ distribution at different globally distributed locations. This suggests that the calibrations and validations with respect to the in-situ references (which are limited to the subtropics) have global validity.

While NDACC/FTIR provides long-term data records, which are very important for climatological studies, MetOp/IASI offers high horizontal resolution, on quasi global scale and morning as well as evening observations. We discuss examples of seasonal cycles as obtained from the NDACC/FTIR $H_2O$-$\delta D$ climatology and demonstrate the potential of IASI for detecting the diurnal variability in the moisture pathways using the summertime Sahara as exemplary area. The few examples reveal that the validated and well-characterised ground- and space-based remote sensing $H_2O$-$\delta D$ pairs can make a significant contribution for investigating moisture transport pathways on different scales.





**Appendix A: MUSICA in the context of other isotopologue ratio remote sensing data sets**

We would like to remark that the results as shown in this paper are only valid for the MUSICA products. This appendix gives a brief overview on other (non-MUSICA) tropospheric water vapour isotopologue remote sensing products and briefly discusses their differences to the MUSICA products.

**A1    The TCCON XH$_2$O and XHDO data**

The ground-based FTIR water vapour isotopologue products that are made available via the TCCON (www.tccon.caltech.edu/) are fundamentally different from the MUSICA ground-based FTIR isotopologue products.

A TCCON-like product is discussed in Rokotyan et al. (2014). It relies on near infrared absorption lines (where HD$^{16}$O is a rather weak absorber) and the isotopologue ratios are calculated aposteriori from independently retrieved H$_2^{16}$O and HD$^{16}$O column amounts. Such aposteriori calculated ratios are affected by the different sensitivities of the individual H$_2^{16}$O and HD$^{16}$O retrievals. At the moment the TCCON kernels do not inform about the cross correlations between the H$_2^{16}$O and HD$^{16}$O product and it is not possible to calculate kernels for humidity and $\delta$D proxies. This means that no Type 2 product can be calculated. The TCCON retrievals use NCEP (National Centers for Environmental Prediction) humidity analyses as H$_2$O apriori and construct the $\delta$D apriori profiles by assuming a fixed linear correlation between $\ln[\mathrm{H_2O}]$ and $\delta$D ($\delta\mathrm{D} = 0.0695 \times \ln[\mathrm{H_2O}] + 0.28$).

**A2    Satellite-based tropospheric water vapour isotopologue data**

A brief overview of available products of tropospheric water vapour isotopologues and the respective satellite sensors is given in the Tables 6 and 7.

The thermal nadir sensors TES and IASI have best sensitivity with respect to the water vapour isotopologues in the middle troposphere (about 2-8 km altitude). In addition to the MUSICA research team a group at the University of Brussels (ULB) is working on IASI water vapour isotopologue retrievals. The ULB IASI retrieval uses two small spectral microwindows (1193-1223 cm$^{-1}$ and 1251-1253 cm$^{-1}$) and fits the proxies for humidity, $\delta$D, and CH$_4$ below 10 km altitude as well as ground temperature (Lacour et al., 2012). It uses the EUMETSAT level 2 temperature output for the whole atmosphere and the EUMETSAT level 2 humidity output for altitudes above 10 km (no fit). The ULB group uses the same globally uniform apriori data as the MUSICA group. For the ULB retrieval, Type 2 (optimally estimated H$_2$O-$\delta$D pair) and Type 1 (individual optimal estimation of H$_2$O and $\delta$D) products can be made available.

Another tropospheric isotopologue product is generated from AURA/TES spectra. It has first been presented by Worden et al. (2006) and the nowadays used TES version 5 product is discussed in Worden et al. (2012). The respective retrieval setup is rather similar to the MUSICA IASI retrieval setup: broad spectral window, simultaneous fit of proxies for humidity and $\delta$D as well as of temperature and the interfering gases throughout the atmosphere. However, for the TES retrieval the H$_2$O apriori assumption comes from the NCEP humidity analyses and the $\delta$D apriori has a latitudinal dependency. The generation of a Type 2 product is theoretically possible, but it is currently not provided. TES measures limb and thermal nadir spectra (the





isotopologue data are generated from the nadir spectra). It has a similar spectral coverage as IASI, but significantly higher spectral resolution and, on the other hand, much sparser daily horizontal coverage.

Space-based sensors measuring solar short wave infrared spectra (SWIR) reflected on the Earth's surface have theoretically better sensitivity in the lower troposphere than the thermal nadir sensors. Retrievals using the sensors SCIAMACHY and

GOSAT have been presented and assessed by Frankenberg et al. (2009), Frankenberg et al. (2013), Boesch et al. (2013), and Scheepmaker et al. (2015). All use humidity analyses (NCEP or ECMWF) as humidity apriori, but a globally uniform $\delta$D apriori. The respective retrievals work independently for the different isotopologues and the isotopologue ratio is calculated after the retrieval process. This is an important difference to the thermal nadir retrievals, which optimally estimate the proxies of $H_2O$ and $HDO/H_2O$. The near infrared retrievals are thus affected by the different sensitivities for the different isotopologues. A

further difference is that in the near infrared the absorption signatures of the secondary isotopologue ($HD^{16}O$) is significantly smaller than in the thermal infrared. The daily horizontal coverage of these sensors is much sparser than for IASI. For the current SWIR retrievals no humidity and $\delta$D proxy kernels are available and it is not possible to assess the difference between the $H_2O$ and $\delta$D kernels and to correct for it, i.e. it is not possible to perform an aposteriori correction and generate a Type 2 product.

## Appendix B:  Modification of HITRAN 2012 line parameter

We allow for a speed-dependent Voigt line shape, whereby we assume a $\Gamma_2/\Gamma_0$ of 15%, which is in good agreement with previous studies (e.g., D'Eu et al., 2002; Schneider et al., 2011) and fit line intensity ($S$) and pressure broadening ($\gamma_{air}$). First, we use the in-situ profiles for empirically estimating the overall errors in $S$ and $\gamma_{air}$ and second, we use the FTIR spectra measured at the three distinct sites (Izaña, Karlsruhe, and Kiruna) for eliminating inconsistencies between the parameters of

the different lines. The theory, practise, and limitations of such empirical line parameter optimisation method are discussed in Schneider and Hase (2009) and Schneider et al. (2011).

Table 8 resumes the modifications we had to make on the HITRAN 2012 parameters for the different lines of Fig. 1 in order to adjust them for a speed-dependent Voigt line shape and for bringing them into agreement with the coincident ISOWAT profile measurements and for minimising the residuals in the spectral fits at Izaña, Karlsruhe, and Kiruna. The obtained values

are in agreement with our previous studies (Schneider and Hase, 2009; Schneider et al., 2011) and they are reasonable in the sense that they lie within the uncertainty ranges as given in the HITRAN data files. A value for $\Gamma_2/\Gamma_0$ of 15% means a line narrowing, which in a Voigt line shape model could be approximated by reducing $\gamma_{air}$ by 4%. In order to counterbalance, the $\gamma_{air}$ parameter had to be generally increased (see last column in Table 8).





## Appendix C: Reference profiles

### C1 Coincidences

During July 2013 we performed an aircraft campaign in the surroundings of Tenerife Island. We operated the ISOWAT instrument (Dyroff et al., 2010) aboard the aircraft and measured highly-resolved vertical profiles of $H_2O$ and $\delta D$ from sea surface

up to almost 7 km altitude on six individual days (130721, 130722, 130724, 130725, 130730, and 130731, Dyroff et al., 2015; Schneider et al., 2015). Figure 11 shows a site map indicating the horizontal flight track of the aircraft (grey line) as well as the location of the Tenerife FTIR instrument (green star) and the IASI observation pixels (coloured squares and diamonds).

The aircraft measures in the free troposphere and the FTIR is based on the island of Tenerife. González et al. (2015) shows that $H_2O$ and $\delta D$ signals as measured in the later morning are already strongly affected by the upslope airflow that is developing

during the day and that early morning data are better representative for the free troposphere. For this reason in Sect. 3 (Fig. 2) we compare FTIR data measured between 8:15 UT and 9:45 UT (solar elevation between 25 and 45°, about 2 and 3.5 hours after sunrise, respectively) with the free tropospheric aircraft measurements, that typically took place between 10:30 and 13:30 UT. This means that there is a temporal mismatch between the FTIR and the ISOWAT observations of up to 5 hours, which is very likely the reason for part of the scatter we observe between the two data sets. Nevertheless, this is what we define as optimal

coincidences. Figure 12 shows the same as Fig. 2 but for FTIR data that have been measured during the time of the aircraft profile measurements, i.e. in the late morning hours and during midday. The comparison plots show relatively large variability in FTIR data that represent the atmospheric layer just above the island, revealing the impact of the local diurnal upslope flow on the FTIR observations. Although for this comparison the temporal mismatches are rather low, it does not represent optimal coincidences, because the strong local upslope flow on the island means that the FTIR and the ISOWAT instrument detect

different air masses.

Tables 9 and 10 list the results of a bias assessment using FTIR observations made for optimal temporal coincidence with the aircraft observations. This assessment leads to different results than the assessment for optimal coincidences (see Tables 9 and 10), thereby revealing the importance for a detailed analysis of optimal coincidences.

The coloured squares and diamonds in Fig. 11 indicate the locations of the IASI observation pixels, whereby the different

colours correspond to the different days (see legend in Fig. 3). We require as coincidence criteria that the observation pixel is not farer away than 50 km from the aircraft's track (coloured squares and diamonds group around the track, which is indicated as the grey line). On the two days 130721 and 130722 there are no IASI pixels within 50 km of the aircraft's track and for those days we also include observation pixels that are located more than 100 km away from the flight track. These are the grey and red pixels marked by a black edge, indicating that they correspond to non-optimal coincidences. In addition there

are three magenta-coloured pixels (representing day 130731) that are marked by black edges. These are also representative for non-optimal coincidences, since on day 130731 there was a very sharp gradient from southeast of the flight track (air mass with strong SAL conditions) to northwest of the flight track (air mass with weaker SAL conditions) and the aircraft's ISOWAT measurements and IASI detect air masses of different characteristics (for a more detailed discussion of this day 130731 please see Appendix A of Schneider et al., 2015).





In Table 10 we resume the results of a MetOp/IASI bias assessment that does not analyse in detail the validity of coincidences (includes the non-optimal coincidences discussed above). The assessed bias values are different to the values as listed in Table 3 manifesting the importance of performing a detailed analyses of the coincidences.

## C2   Ceiling altitude and uncertainties

The basis of the references are the ISOWAT measurements (surface up to almost 7 km) and the climatological values assumed above the ceiling altitude (see Sect. 2.3.3). This profile (ISOWAT + climatology) is then smoothed by the averaging kernel of the remote sensor. The uncertainty of ISOWAT is 4% for $H_2O$ and typically better than 10‰ for $\delta D$ (only for rather dry air it can reach high values, e.g. 35‰ when the $H_2O$ concentration is below 500 ppm, Dyroff et al., 2015). However, above the ceiling altitude we have no measurements and the uncertainty is significantly larger. We assume 100% and 80‰ for $H_2O$ and

$\delta D$, respectively. These large uncertainties for the atmosphere above the ceiling altitude propagate to lower altitudes due to the smoothing with the averaging kernels. In fact, the error bars on the reference data for 4.9 km as well as for 2.4 km as depicted in Figs. 2, 3 and 12 are mainly due to the unknown $H_2O$ and $\delta D$ values above 7 km.

For the references used for the FTIR data validation at 2.4 km altitude (top panels of Figs. 2 and 12) the uncertainty introduced from missing data above the ceiling altitude is about 15% for $H_2O$ and 12‰ for $\delta D$. For the FTIR data validation at

4.9 km altitude (bottom panels of Figs. 2 and 12) the respective uncertainty is about 25% for $H_2O$ and 20‰ for $\delta D$. For the IASI data validation at 4.9 km the respective uncertainties are 25%-6% for $H_2O$ and 20-5‰ for $\delta D$. The values depend on both the averaging kernels as well as the ceiling altitude (e.g. on day 130721 we only reached 6.0 km leading to higher uncertainties than for the other days when we typically reached 6.8 km).

For the validation/calibration of the remote sensing data it is essential to have reference measurements that cover the tropo-

sphere from the surface up to high altitudes. During the ISOWAT campaign we almost reached 7 km during most of the flights and we are not aware of another data set with similarly good altitude coverage.

## Appendix D:  Daily coverage for IASI Type 1 product

In Sects. 6 and 4.2.1 we present IASI Type 2 products. For details about the Type 1 and 2 classification please refer to Schneider et al. (2012) and Wiegele et al. (2014). For Type 2 the remote sensing $H_2O$ and $\delta D$ data are representative for

the same air mass, which allows studies with $H_2O$-$\delta D$ plots. However, actually the remote sensors are much more sensitive for $H_2O$ than for $\delta D$. The Type 1 product consists of vertical $H_2O$ profiles and of $\delta D$ values that are representative for the middle troposphere only. This makes an adequate interpretation of $H_2O$-$\delta D$ plots almost impossible. However, in case a user is only interested in $H_2O$ and not in the isotopologue, the Type 1 product is recommendable.

In Fig. 13 we depict $H_2O$ maps as obtained from morning overpasses on a winter day (upper panel) and on a summer day

(bottom panel). We only plot data for which we have a DOFS (degree of freedom for signal) for $H_2O$ of at least 3.0. If we compare these maps with the maps of Figs. 8 and 9 we see that we can often detect good $H_2O$ profiles at locations where we do not have sufficient sensitivity for the middle tropospheric isotopologue ratios.





*Acknowledgements.* This study has been conducted in the framework of the project MUSICA which is funded by the European Research Council under the European Community's Seventh Framework Programme (FP7/2007-2013) / ERC Grant agreement number 256961.

E. Sepúlveda is supported by the Ministerio de Economía and Competitividad of Spain for the project NOVIA (CGL2012-37505).

The aircraft campaign has been co-funded by the project MUSICA and the Spanish national project AMISOC (CGL2011-24891).

We are grateful to INTA Aerial Platforms, a branch of the Spanish ICTS program, and the Spanish Air Force for their efforts in maintaining and operating the aircraft.

The AERONET sun photometer at Izaña (PI: Dr. Emilio Cuevas) has been calibrated within AERONET EUROPE TNA supported by the European Community Research Infrastructure Action under the FP7 Capacities program for Integrating Activities, ACTRIS grant agreement number 262254.

The Izaña aerosol in-situ measurements are part of the project POLLINDUST (CGL2011-26259) funded by the Minister of Economy and Competitiveness of Spain.

We thank all the personal from the Izaña Atmospheric Research Center (IARC) of the Agencia Estatal de Meteorología (AEMET). Our study has strongly benefitted from this great support and important measurements have been made in IARC research facilities.

We acknowledge the support by the Deutsche Forschungsgemeinschaft and the Open Access Publishing Fund of the Karlsruhe Institute

of Technology.

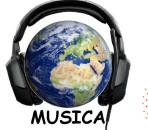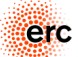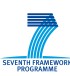





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



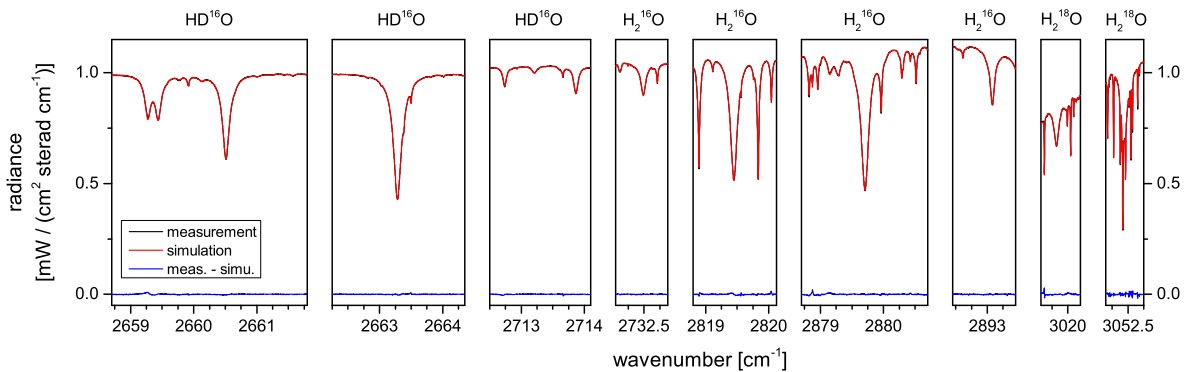

**Figure 1.** The spectral windows used for the version 2015 of the MUSICA ground-based NDACC/FTIR retrievals. Shown is an example for a typical measurement at Izaña (26 October 2011, 11:02 UT; solar elevation: 41.7°; $H_2O$ slant column: 6.3 mm). Black line: measurement; red dashed line: simulation; blue line: residual (difference between measurement and simulation).





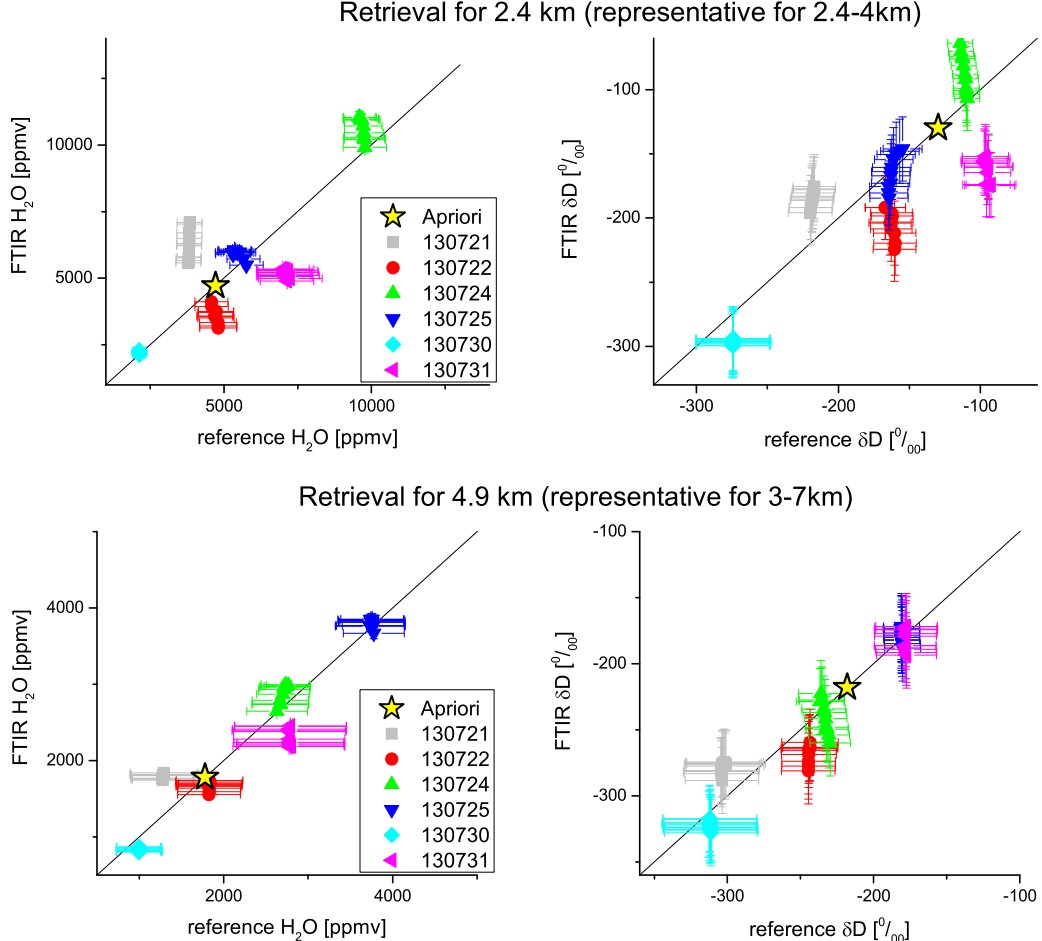

**Figure 2.** Correlation between the reference profiles (in-situ data measured by the ISOWAT instrument for 0-7 km and climatology above, then smoothed with FTIR kernels) and FTIR data for FTIR measurements made in the morning (8:15 - 9:45 UT), i.e. before the aircraft flights, but reasonable representative for the free troposphere. Left panels for $H_2O$ and right panels for $\delta D$. Upper panels for retrievals at 2.4 km and bottom panels for retrievals at 5 km. The black line is the 1:1 diagonal. The error bars represent the uncertainty estimations for the reference and FTIR data. Note that for the comparison at 5 km a large part of the uncertainty in the reference data is due to the fact that there are no ISOWAT measurements above the aircraft's ceiling altitude (Schneider et al., 2015, and discussion in Appendix C).





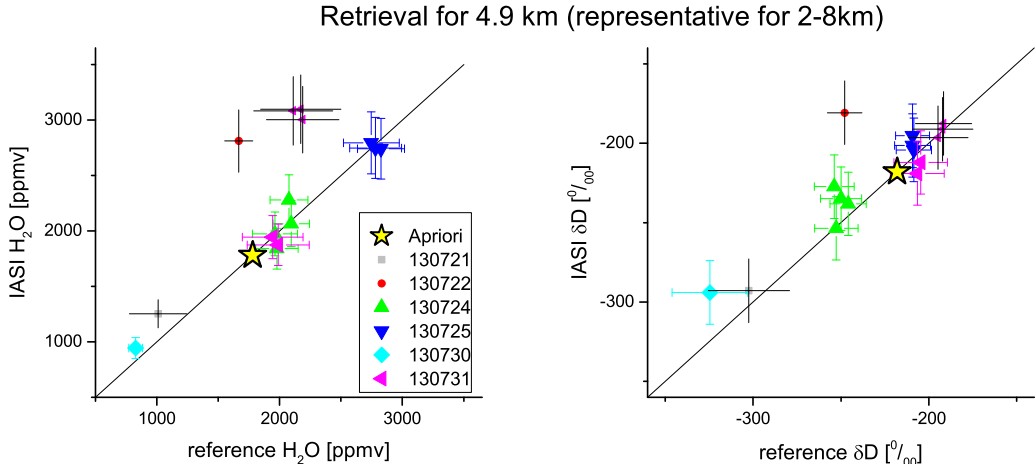

**Figure 3.** Same as Fig. 2 but for the correlation between reference data (in-situ data measured by the ISOWAT instrument for 0-7 km and climatology above, then smoothed with IASI kernels) and IASI data for 5 km altitude. The colour code is for the different days. The data points for non-optimal coincidences as discussed in Appendix C can be identified by the smaller symbols and the black error bars.



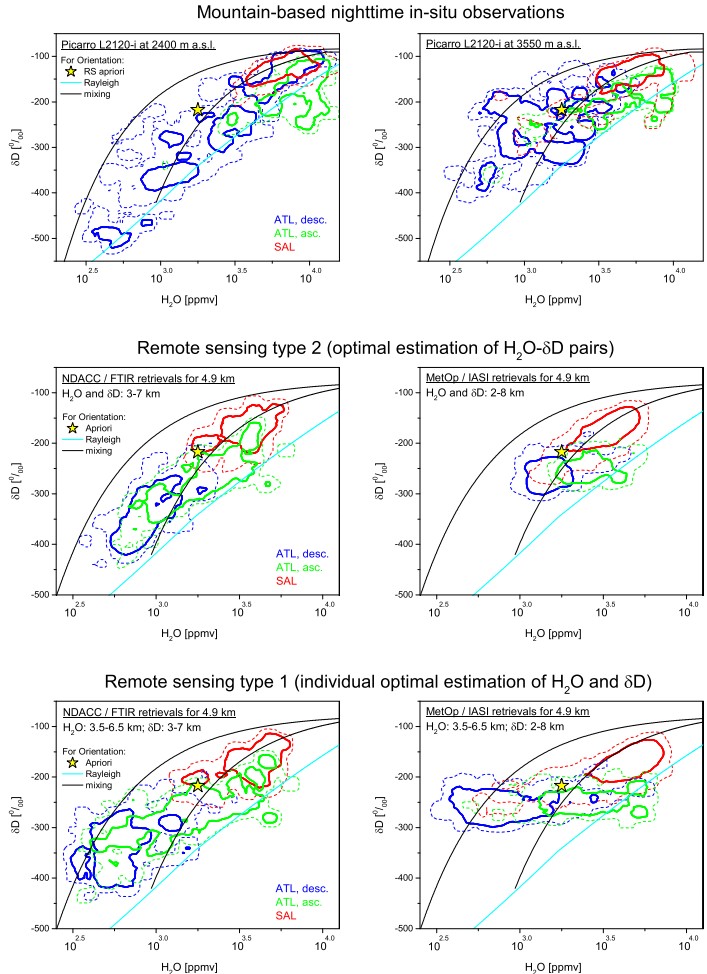

**Figure 4.** $H_2O$-$\delta D$ distribution as obtained for the free troposphere in the surroundings of Tenerife. Upper panel: in-situ reference; Middle panels: remote sensing Type 2 product (Left: ground-based NDACC/FTIR located on Tenerife at the Izaña observatory (2370 m a.s.l.); Right: space-based MetOp/IASI-A and -B observing in a $2° \times 2°$ area around Tenerife); Bottom panels: Same as middle panels but for the remote sensing Type 1 products. The contour lines indicate areas of highest densities: Red for air masses that are clearly affected by dry convection over the African continent; Blue and green for Atlantic air masses with different pathways (see Table4). The thin dashed and thick solid lines mark the areas that include 95% and 66% of all data, respectively. In addition, the panels show three simulated curves: a Rayleigh curve for initialisation with $T = 25°C$, $RH = 80\%$ and $\delta D = -80‰$ (cyan) and two mixing curves (black, first line for mixing between $H_2O[1] = 25000$ ppmv; $\delta D[1] = -80‰$ and $H_2O[2] = 900$ ppmv; $\delta D[2] = -430‰$ and second line for mixing between $H_2O[1] = 25000$ ppmv; $\delta D[1] = -80‰$ and $H_2O[2] = 200$ ppmv; $\delta D[2] = -610‰$). The yellow star marks the apriori value used for the remote sensing retrievals at 4.9 km.





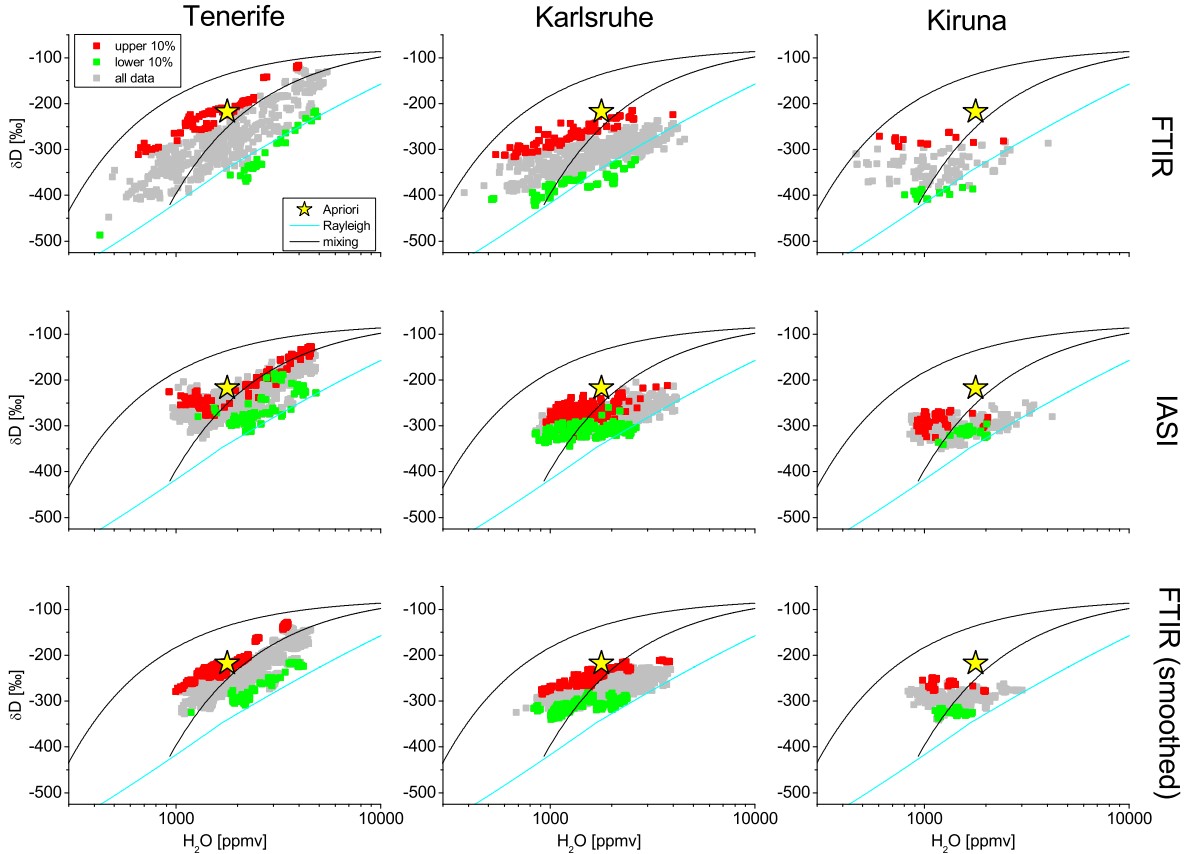

**Figure 5.** $H_2O$-$\delta D$ plots for 5 km altitude for coincident FTIR and IASI measurements for the three locations Tenerife, Karlsruhe, and Kiruna. Plotted are (from the top to the bottom) the FTIR data, the IASI data, and FTIR data smoothed with the IASI averaging kernels. The colour code displays the upper 10% and lower 10% of $\delta D$ values as identified in the FTIR data. The cyan line, the black lines and the yellow stars are the same as in Fig.4 (Rayleigh line, mixing lines and apriori value for 5 km altitude, respectively). For more details on this kind of validation approach, please refer to Wiegele et al. (2014).



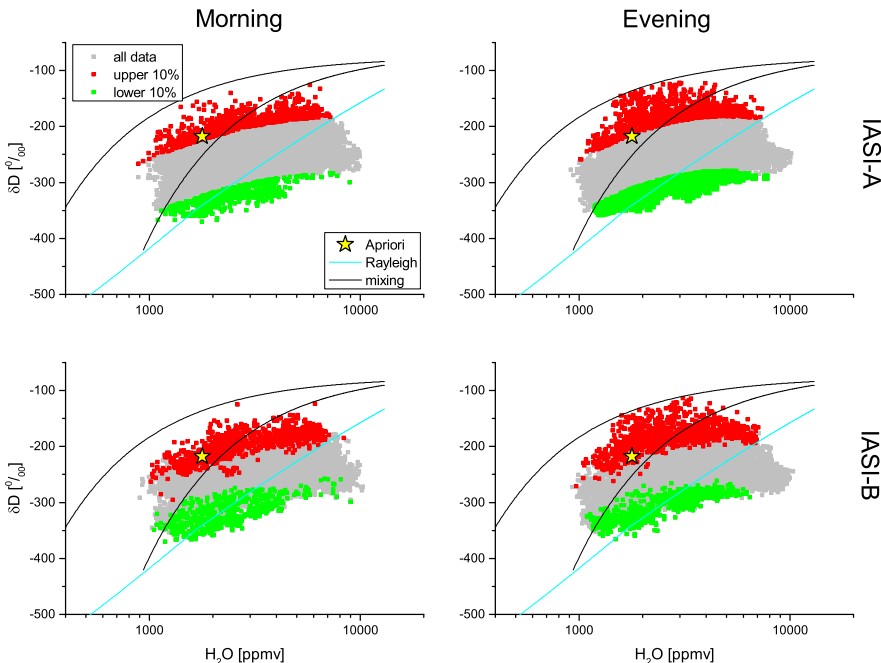

**Figure 6.** Similar to Fig. 5 but for coincidences of IASI-A and IASI-B measurements. Presented are the products retrieved at 5 km for all coincidences within 1 hour and $0.25° \times 0.25°$ for 16 August 2014. The left panels show the morning overpasses and right panels the evening overpasses. The anomalies are identified in the IASI-A observations (upper panels) and then checked in the IASI-B observations (bottom panels).





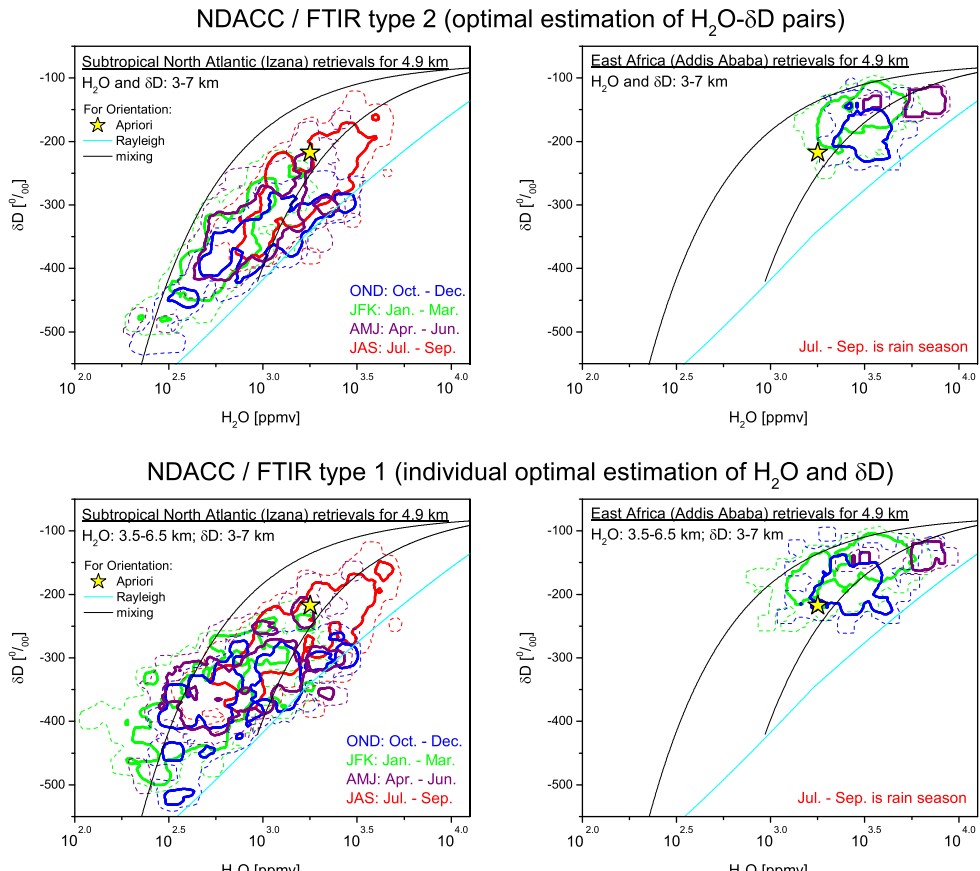

**Figure 7.** Example of seasonal climatologies obtained from FTIR retrievals at Izaña/Tenerife (left panels) and Addis Ababa (right panels). Top panels are for Type 2 and bottom panels are for Type 1 products. October-December: blue contour lines; January-March: green contour lines; April-June: purple contour lines; July-September: red contour lines. The thin dashed and thick solid lines mark the areas that include 95% and 66% of all data, respectively. Yellow star, black line and cyan lines are the apriori and the simulated lines as in Figs. 4 and 5.





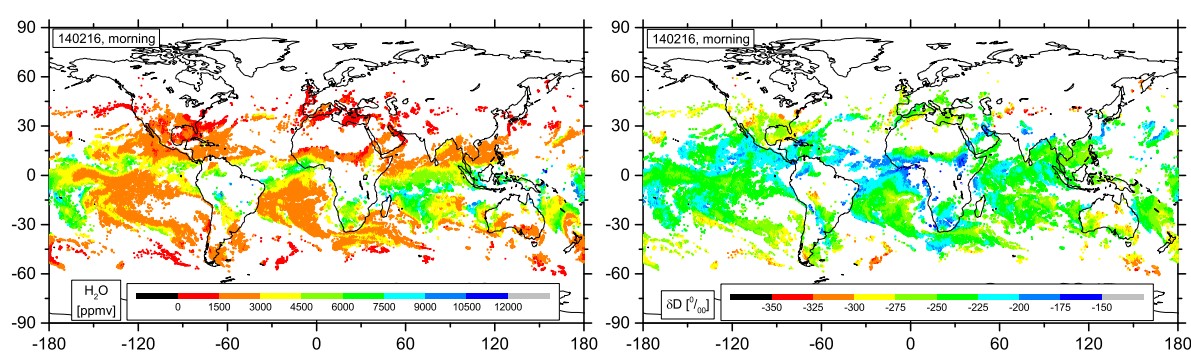

**Figure 8.** Morning overpass map with the MUSICA IASI-A and -B quality filtered Type 2 retrieval results for 5 km altitude (example 16 February 2014). Left for $H_2O$ [ppmv] and right for $\delta D$ [‰].




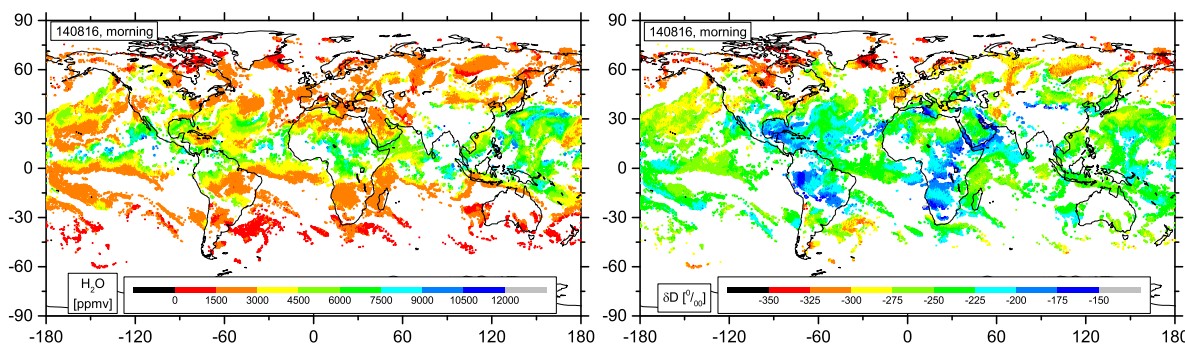

**Figure 9.** Same as Fig. 8, but example for 16 August 2014.





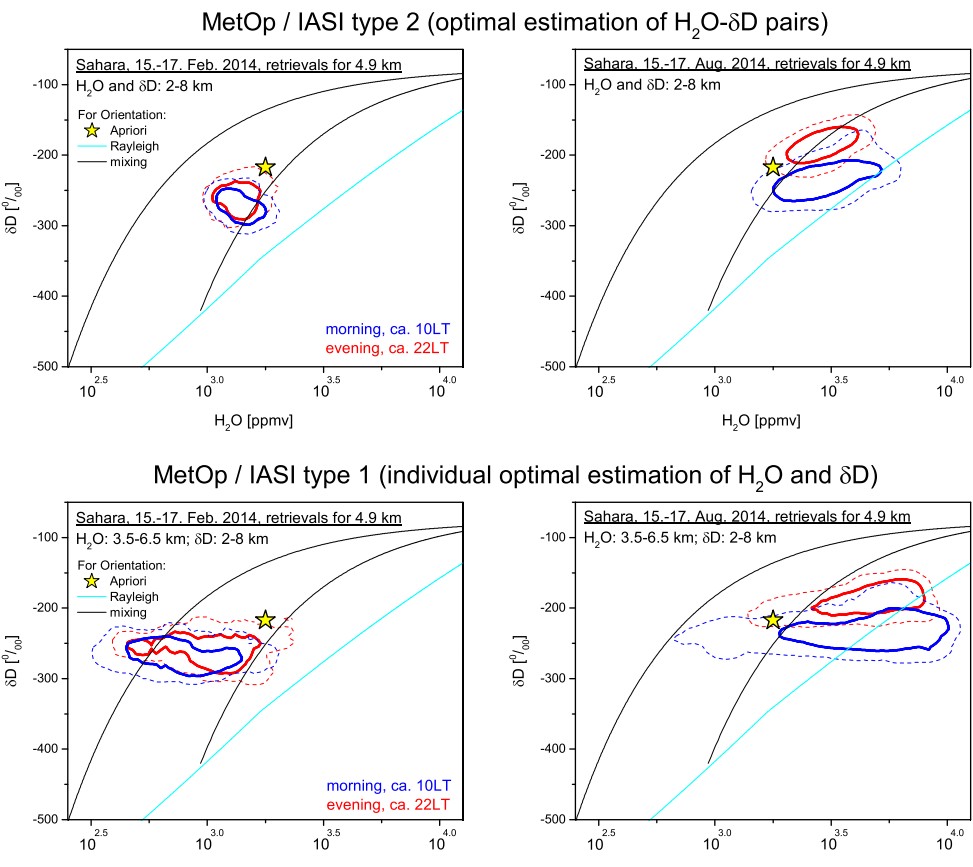

**Figure 10.** Example of diurnal signals in the IASI retrievals of the Sahara desert (22.5 to 32.5°N and 10°W to 30°E). Left panels are for boreal winter days (15.-17. February 2014) and right panels for boreal summer days (15.-17. August 2014). Top panels are for Type 2 and bottom panels are for Type 1 products. Morning data: blue contour lines; Evening data: red contour lines. The thin dashed and thick solid lines mark the areas that include 95% and 66% of all data, respectively. Yellow star, black line and cyan lines are the apriori and the simulated lines as in Figs. 4, 5 and 7.




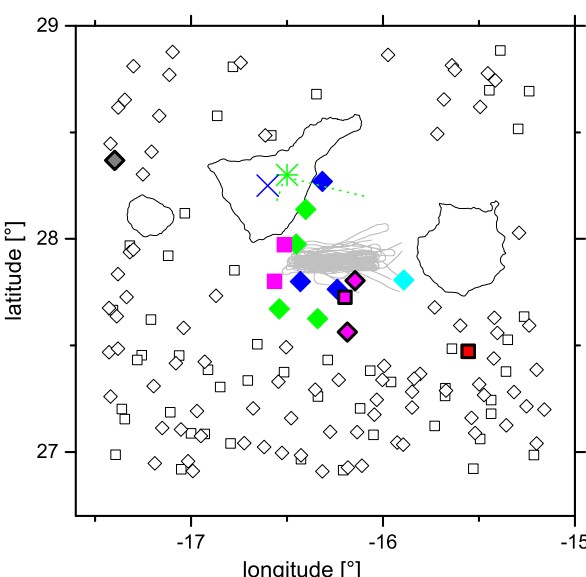

**Figure 11.** Site map indicating the location of the different instruments and ground pixels during the aircraft campaign on six days in July 2013. Green star: Izaña observatory (location of the first Picarro and the FTIR, the green dashed lines indicate the line of sight of the FTIR between 8:15 and 13:30 UT); Blue star: Teide observatory (location of the second Picarro); grey lines: aircraft flight track during ISOWAT measurements; Black squares and diamonds: cloud free ground pixels of IASI-A and -B, respectively, during the six aircraft flights; Red filled squares and diamonds: pixels that fulfill our coincidence criteria for IASI-A and -B, respectively, whereby the different filling colour corresponds to the six different days as in Figs. 2 and 3.



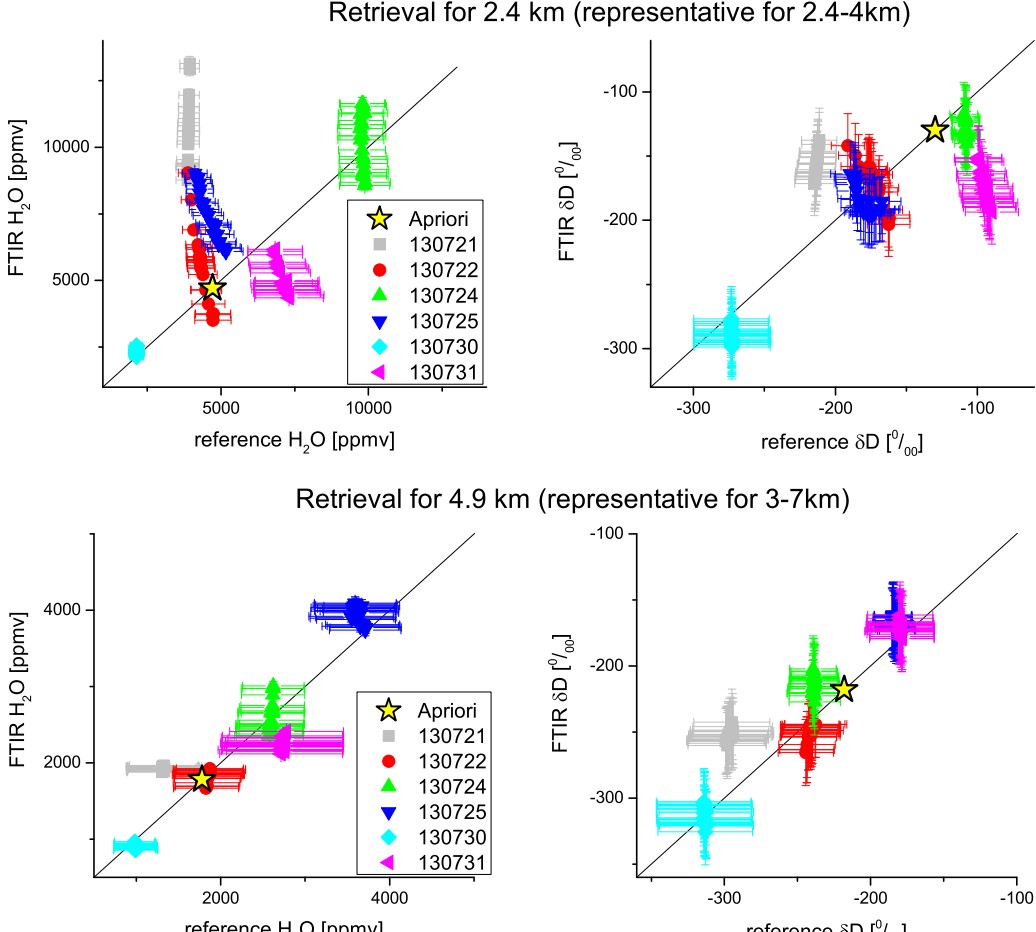

**Figure 12.** Same as Fig. 2, but for FTIR measurements corresponding to best temporal coincidences (FTIR observations made during the 3 hours of the aircraft flights, typically 10:30 - 13:30 UT).





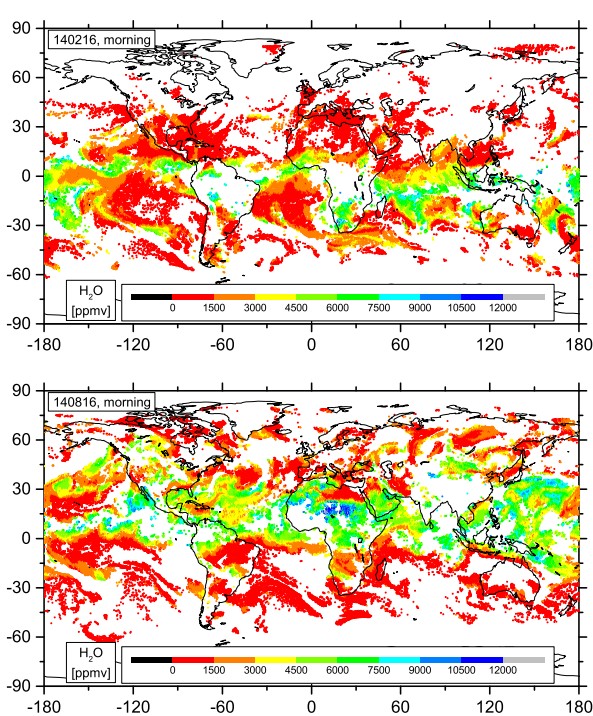

**Figure 13.** Morning overpass map with MUSICA IASI A and B quality filtered Type 1 retrieval results ($H_2O$ profiles, shown are here values for 5 km altitude). Upper panel: example for 16 February 2014; Bottom panel: example for 16 August 2014.





**Table 1.** Important developments/milestones in the context of the MUSICA activities.

| Development/Milestone | References |
| --- | --- |
| Optimal estimation of $H_2O$, HDO and $\delta D$ | Schneider et al. (2006) |
| Improving spectroscopic line parameterisation using atmospheric spectra | Schneider and Hase (2009); Schneider et al. (2011) |
| The MUSICA FTIR/NDACC retrieval, humidity and $\delta D$ proxies | Schneider et al. (2012) |
| Aposteriori processing for generating optimal $H_2O$-$\delta D$ pairs | Schneider et al. (2012) |
| The MUSICA MetOp/IASI retrieval | Schneider and Hase (2011); Wiegele et al. (2014) |
| Validation of $H_2O$-$\delta D$ pairs | Wiegele et al. (2014); Schneider et al. (2015) |
| Using $XCO_2$ for quality filtering of MUSICA NDACC/FTIR | Barthlott et al. (2015) |
| In-situ profile references (ISOWAT aircraft campaign, 0-7 km) | Dyroff et al. (2015) |
| Continuous in-situ reference for the free troposphere | González et al. (2015) |





**Table 2.** Empirical assessment of the bias for the NDACC/FTIR $H_2O$ and $\delta D$ products in the lower troposphere.

| Sensor | Altitude Range | Number of Remote Sensing Observations | Number of Reference Observation ($N_R$) | Mean Bias $\pm$ Confidence $H_2O$ [%] | $\delta D$ [‰] |
|---|---|---|---|---|---|
| NDACC/FTIR | 2.4-4 km | 65 | 6 | $+2.1 \pm 12.4$ | $-12.1 \pm 16.6$ |



**Table 3.** Empirical assessment of the bias for the NDACC/FTIR and MetOp/IASI $H_2O$ and $\delta D$ products in the middle troposphere.

| Sensor | Altitude Range | Number of Remote Sensing Observations | Number of Reference Observation ($N_R$) | Mean Bias $\pm$ Confidence $H_2O$ [%] | $\delta D$ [‰] |
|---|---|---|---|---|---|
| NDACC/FTIR | 3-7 km | 65 | 6 | $-0.8 \pm 8.2$ | $-2.7 \pm 7.4$ |
| MetOp/IASI | 2-8 km | 10 | 4 | $+0.6 \pm 3.7$ | $+8.5 \pm 7.9$ |



**Table 4.** The dominant moisture pathways to the free troposphere in the surroundings of Tenerife Island.

| Pathway | Description | Identification Method |
|---|---|---|
| ATL, desc. | Atlantic air mass descending from high latitudes | low aerosol load (measurement) and low temperature at point of last condensation (trajectories), González et al. (2015) |
| ATL, asc. | Atlantic air mass ascending from lower latitudes | low aerosol load (measurement) and high temperature at point of last condensation (trajectories), González et al. (2015) |
| SAL | Saharan Air Layer advected over the Atlantic | high aerosol load (measurement), Rodríguez et al. (2011); González et al. (2015) |

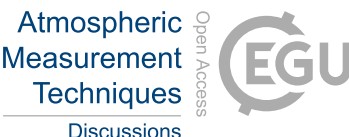

**Table 5.** List of current MUSICA NDACC/FTIR sites (ordered from North to South) and available MUSICA data record.

| Site | Location | Data Record |
|---|---|---|
| Eureka, Canada | 80.1°N, 86.4°W | 2006-2014 |
| Ny Alesund, Norway | 78.9°N, 11.9°E | 2005-2014 |
| Kiruna, Sweden | 67.8°N, 20.4°E | 1996-2014 |
| Bremen, Germany | 53.1°N, 8.9°E | 2004-2014 |
| Karlsruhe, Germany | 49.1°N, 8.4°E | 2010-2014 |
| Jungfraujoch, Switzerland | 46.6°N, 8.0°E | 1996-2014 |
| Izaña/Tenerife, Spain | 28.3°N, 16.5°W | 1999-2014 |
| Altzomoni, Mexico | 19.1°N, 98.7°W | 2012-2014 |
| Addis Ababa, Ethiopia | 9.0°N, 38.8°E | 2009-2013 |
| Wollongong, Australia | 34.5°S, 150.9°E | 2007-2014 |
| Lauder, New Zealand | 45.1°S, 169.7°E | 1997-2014 |
| Arrival Heights, Antarctica | 77.8°S, 166.7°E | 2002-2014 |



**Table 6.** Overview on tropospheric water vapour isotopologue retrievals using space-based observations.

| Research Group / Sensor | Spectral Window | Fitted Parameter | References and Remarks |
|---|---|---|---|
| MUSICA (IMK-ASF) / IASI | $1190\text{-}1400\,\text{cm}^{-1}$ | $(\ln[H_2O] + \ln[HDO])/2$<br>$\ln[HDO] - \ln[H_2O]$<br>$CH_4$, $N_2O$, $CO_2$, and $HNO_3$<br>atmospheric temperature<br>ground temperature | Schneider and Hase (2011), Wiegele et al. (2014),<br>only clear sky retrievals |
| ULB (U. Brussels) / IASI | $1193\text{-}1223\,\text{cm}^{-1}$<br>$1251\text{-}1253\,\text{cm}^{-1}$ | $(\ln[H_2O] + \ln[HDO])/2$<br>$\ln[HDO] - \ln[H_2O]$<br>$CH_4$<br>ground temperature | Lacour et al. (2012),<br>only fit for 0-10 km,<br>atm. temp. from EUMETSAT L2,<br>only clear sky retrievals |
| NASA / TES | $1170\text{-}1320\,\text{cm}^{-1}$ | $(\ln[H_2O] + \ln[HDO])/2$<br>$\ln[HDO] - \ln[H_2O]$<br>$CH_4$ and $N_2O$<br>atmospheric temperature<br>ground temperature<br>cloud ($\tau$ and pressure) | Worden et al. (2006), Worden et al. (2012) |
| SRON / SCIAMACHY | $4212\text{-}4248\,\text{cm}^{-1}$ | $H_2^{16}O$, $H_2^{18}O$, and $HD^{16}O$<br>$CH_4$ and CO | Frankenberg et al. (2009),<br>Scheepmaker et al. (2015) |
| NASA / GOSAT | $6311\text{-}6441\,\text{cm}^{-1}$ | $H_2O$ and HDO | Frankenberg et al. (2013) |
| U. Leicester / GOSAT | $6439\text{-}6464\,\text{cm}^{-1}$ | $H_2O$ and HDO | Boesch et al. (2013),<br>uses $CH_4$ obtained from extra $CH_4$ retrieval |





**Table 7.** Space-based sensors with available tropospheric water vapour isotopologue retrieval products.

| Sensor | Meas. Geo. | Pixel Size | Meas. per day | Temporal Coverage | Spectral Res. |
|--------|-----------|-----------|---------------|-------------------|---------------|
| IASI | thermal nadir | 12 km diameter (at nadir) | ≈ 1.3 Mio. | IASI-A: since 2007 IASI-B: since 2013 IASI-C: scheduled for 2018 | $0.5\,cm^{-1}$ |
| TES | thermal nadir | 5×8 km (at nadir) | ≈ 2100 | since 2002 (since 2010 temporarily) | $0.1\,cm^{-1}$ |
| SCIAMACHY | SWIR | 120×30 km | ≈ 32000 | 2003-2012 (after 2007 increased detector degradation) | $0.45\,cm^{-1}$ $0.45\,cm^{-1}$ |
| GOSAT | SWIR | 10 km diameter | ≈ 10000 | since 2009 | $0.4\,cm^{-1}$ |




**Table 8.** Modifications in the line parameters (line intensity and pressure broadening) made with respect to HITRAN 2012.

| Line Centre [cm$^{-1}$] | Isotopologue | $\Delta S$ [%] | $\Delta\gamma$ [%] |
|---|---|---|---|
| 2660.511700 | HD$^{16}$O | -5.52 | +3.96 |
| 2663.285820 | HD$^{16}$O | -5.53 | +4.00 |
| 2713.862650 | HD$^{16}$O | -5.53 | +4.07 |
| 2732.493160 | H$_2^{16}$O | +12.26 | +9.35 |
| 2819.449040 | H$_2^{16}$O | -3.07 | +4.52 |
| 2879.706660 | H$_2^{16}$O | -8.26 | +6.84 |
| 2893.075920 | H$_2^{16}$O | -9.07 | +9.64 |
| 3019.824500 | H$_2^{18}$O | -5.40 | -0.72 |
| 3052.444870 | H$_2^{18}$O | -6.32 | -0.71 |



**Table 9.** Same as Table 2, but for all best temporal coincidences (which are not necessarily optimal coincidences).

| Sensor | Altitude Range | Number of Remote Sensing Observations | Number of Reference Observation ($N_R$) | Mean Bias ± Confidence | |
|---|---|---|---|---|---|
| | | | | $H_2O$ [%] | $\delta D$ [‰] |
| NDACC/FTIR | 2.4-4 km | 142 | 6 | $+24.5 \pm 19.7$ | $-10.23 \pm 19.2$ |



**Table 10.** Same as Table 3, but for all best temporal coincidences (which are not necessarily optimal coincidences).

| Sensor | Altitude Range | Number of Remote Sensing Observations | Number of Reference Observation ($N_R$) | Mean Bias ± Confidence | |
|---|---|---|---|---|---|
| | | | | $H_2O$ [%] | $\delta D$ [‰] |
| NDACC/FTIR | 3-7 km | 142 | 6 | +2.7 ± 8.3 | +14.0 ± 8.2 |
| MetOp/IASI | 2-8 km | 15 | 6 | +12.3 ± 8.4 | +11.0 ± 8.6 |