# Peer review of "A framework for accurate, long-term, global and high resolution observations of tropospheric H$_2$O-$\delta$D pairs — a MUSICA review"

_Atmospheric Measurement Techniques, 2015_

## Referee Comment (RC1) · Anonymous Referee #2 · 17 Feb 2016

The paper "A framework for accurate, long-term, global and high resolution observations of tropospheric H2O-$\delta$D pairs - a MUSICA review" presents the results of the MUSICA project which aims at providing coherent $\delta$D-H2O observations from different platforms. While no new scientific progress is reported, the publication of this paper as a review paper makes sense for potential users of $\delta$D because of the complex nature of such remote sensed products. Moreover a summary paper on the different available products would nicely help the potential users. I would however strongly recommend to the authors to make changes and cuts in order to make it more like a review paper. Some sections provide redundant information which was already provided in your previous papers. It would be nice to illustrate the long term aspect of your FTIR data by

showing long term time series. It would also be useful for potential users to document the capabilities of all the FTIR stations in term of vertical sensitivity (averaging kernels and DOFS, I did not find this information in your previous papers).

**General comments:**

Section 2 - Retrieval description:

You spend some time describing the retrieval setup, it would be also useful to report the noise levels you used for the FTIR and IASI retrievals. Also you state that you fit the temperature from IASI spectra (section 2.3.2), is it the same procedure than in Wiegele et al., 2014 (a priori is the EUMETSAT L2 with a strong constraint)? In that case it would be interesting to report it as well because it means you are sensitive to EUMETSAT temperature errors.

Section 3: Accurate remote sensing products

- The use of accurate here suggests that there is no error in the retrievals, the authors however nicely describe a systematic bias between certain ranges. The description of a most likely range of bias is very useful for the users but it also indicates that the retrievals are not exact like the title suggests. I recommend to modify the title to a less overselling one, such as "absolute characterization of remote sensing products".

- Could you provide the numbers in the description of the correlation (correlation coefficient and associated p-values). Section 3.1 L18 and Section 3.2 L6

- The comparison of retrieval products with non-smoothed in situ data is also of interest for the users as it allows to realize how much the smoothing affects the data. Could you
add a comparison between non-smoothed data (like you do in Figure 5) and retrieved ones or say a word about how the smoothing affects the in situ data.

Section 4: Validation of H2O-deltaD pairs

It is true validating the added value of deltaD retrieval is very important but it has already been made in Schneider et al., 2015 and Wiegele et al., 2014. What is the interest to present here the moisture pathways and the extreme approaches again? The novelty compared to these previous papers is not clear. Moreover the message of these two approaches is the same. I recommend presenting only one or the other.

Is there a reason why you would expect differences between IASI-A and IASI-B? Are the radiances measurements not calibrated and compared? Could you develop why you need to verify.

Section 5: Consistent long-term observation with NDACC/FTIR

- It would be useful in the perspective of a review paper on the capabilities of the MUSICA network to present the vertical sensitivities of the different FTIR sites. What vertical sensitivity can we expect from the new FTIR? Maybe within the Table 5, you could add a column with the altitude range of sensitivity and the dofs. - An example (or several) of long term times series would nicely illustrate the title of the section. Is there any inter annual variability, do you observe trends? Such questions would greatly help to assess the added value of such long term timeseries. - The lower panel of Figure 7 shouldn't appear in this section, it is more related to section 4 but the message was already clear from section 4.1.

Section 6: Quasi global and high resolution observations with IASI

- The processing of all IASI data is very challenging and requires the development of dedicated solution, are you able to process the IASI data in near real time? It is also an information that data users are interested in and that would fit within this section.
- Figure 8 and 9 are redundant, one is enough to illustrate your point. - The type 1 versus type 2 discussion would better fit in section 4.

Appendix D

This appendix could be summarized in the article in one sentence without associated figure. I understand that the coverage is slightly better for type I product but I believe one number would be enough to illustrate that.

**Minor comments**

- "Exemplary": often used throughout the manuscript, if possible, alternate

- P2, L14: deltaD x 1000

- P2 L31: "for investigating the dynamics of the MJO (..)", add Tuinenburg et al., 2015 as a reference concerning the study of MJO with IASI.

- P2, L33: "However, there is only one study so far where attempts have been made for empirically validating these H2O-$\delta$D pairs (Schneider et al.,2015)" – Herman et al., 2014 and Worden et al., 2011 are previous attempts to validate TES deltaD retrievals, please consider them in this sentence.
[Figure]

- P2, L35: "Further and more detailed validation efforts for the H2O-$\delta$D pairs are urgently needed"- Urgently sounds a bit over dramatic regarding the numerous publications that have been made since the TES data (2006) for example.

- P5, L24: "The Type 2 product offers consistent H2O-$\delta$D pairs, which are sensitive to the lower and the middle troposphere, whereby it is possible to reasonably separate both altitude regions (degree of freedom for signal, DOFS, is typically about 1.8)." – Is it the case for all FTIR sites? What are the dofs for the different sites? This information would be useful in the table 5.

- P7, L2: "We would like to remind that there is no similar study where the accuracy of simultaneous tropospheric H2O and $\delta$D remote sensing observations is documented by such direct comparison to coincident reference profiles". – What about Herman et al., 2014?

- P7, L18: Please specify the coefficient correlation values

- P8, L2: The figure mentioned here shows spatial colocation of the different measurements and does not give any information on the bias. The figure is not needed in this context, referencing to Schneider et al (2015) is sufficient.

- P12,L18-22: "the MUSICA NDACC/FTIR isotopologue data are representative for different altitudes and for larger scale processes (the data represent vertical layers averaged over 2-5 km (see typical averaging kernels in Fig. 3 of Schneider et al., 2015) "- Again, are these "typical" averaging kernels representative of all the FTIR stations? A large scale process better defines a process that occurs at a wide horizontal scale than a vertical one. I suggest using "deeper layers".

- P13, L4:10: " After the rain season (OND, blue) the vapour is most depleted in HDO. Then evaporation of precipitated surface water might be more important as free tropospheric moisture source than during the other seasons. Between January and March (JAM, green) the vapour concentrations remain similar as during

the previous months, but the air becomes more enriched in HDO, which might indicate increased importance of mixing with humid boundary layer air and reduced importance of condensation processes. For April to June (AMJ, purple) air gets more humid, but $\delta D$ remains almost constant, indicating to further moistening due to even stronger mixing with humid boundary layer air." - The interpretation of $\delta D$ is not straightforward and several combinations of different processes can explain a same H2O- $\delta D$ composition. I would keep this part very brief to avoid erroneous interpretations. For example you suggest that the surface evaporation in OND will give water vapor a more depleted signature. However, evaporation from surface tends to enrich water vapor as there is no fractionation during evapotranspiration. Or in MAM, different airmass trajectories from different sources with same mixing historic could explain the observed signatures.

- P13, L15: "The IASI ground pixel at nadir has a diameter of only 12 km" – 12 km is not that small, GOSAT as TES ground pixels are smaller, moreover there is less nadir measurement than not nadir measurement because of the large IASI swath, remove only.

- P13,L23: "Please recall that one data point represents a ground pixel of 12 km diameter (at nadir)" – This sentence is not necessary as it was said a few lines before.

- P17,L2: IASI and TES retrievals have been cross-validated in Lacour et al., 2015 you should add this reference here. Also add the reference in the Table 6.

---

## Referee Comment (RC2) · Anonymous Referee #1 · 22 Feb 2016

The manuscript AMT-2015-330 is a review of the development of the MUSICA framework for observing H2O-delta_D pairs and a report of its current state. The authors describe very detailed their solutions of the lacks in the calibration and validation of the retrievals, partially published already in previous articles. The manuscript is within the scope of the journal AMT. A review of the evolution in this intricate research field merits publication. But, as this being a review article, I expect the content being presented in a more general context, and, maybe with less details of the own work, that has been published before.

General comments:

1. Introduction:

[Figure]

A more general introduction into the research field of atmospheric water vapour iso-topologues within the context of atmospheric dynamics should be added. The introduction currently reads more as an excuse for the authors work. A wider overview should introduce the reader into this topic and should set the authors work to less profile. This means also, the content of Appendix A should be shifted into the introduction, while Appendix A could be erased. There was previous work published (e.g. Worden et al.,ACP , 2011). This should be cited correctly. The authors generally should set their work more into the context. Less noisy advertising would be more satisfactory for a review.

2. Remote sensing of H2O and delta_D:

The authors describe their retrievals in great detail, which is acceptable here. They quantify their obtained errors, but, a general discussion about error sources is missing. This should necessarily be added.

3. Accurate remote sensing products:

This section presents the main results of the development of MUSICA remote sensing products and claims their high quality and accuracy. For evaluating the results, correlation coefficients and slope-values should be added to the tables 2 and 3. A discussion about the limits of the described products is largely missing. Appendix C points out that coincidence of the reference measurements was limited. But the impact of short term and small scale variability of tropospheric humidity is not discussed or even quantified. To my feeling a detailed discussion about this is necessary for any evaluation based on reference data with a temporal and spatial mismatch. Quantitative analyses of tropospheric variability of water vapour have recently been published in ACP (Steinke et al., 2015 and Vogelmann et al. 2015). These studies should be considered and cited.

It is stated, that one error source is the unknown humidity above the ceiling altitude of the aircraft. For me, this seems not to be that clear. Above 7km I would expect definitely less than 5% of the total water vapour column and consequently variations being even

significantly smaller which is in a certain contrast to the named errors. However, I would expect larger uncertainties caused by short term and small scale variability.

4. Validation of H2O and delta_D pairs:

I agree with Referee #2. This section is a somewhat extensive and could be more a summary of the previous work (Wiegele 2014, Schneider 2015).

5. Consistent long-term observation with NDACC/FTIR:

The contour plots of Fig. 7 show, that their is a seasonality in H20-delta_D-pairs, but I would expect at least something like a long time series for different locations. Such a plot should have the time-scale on the x-axis. It might be better, to move the interpretation of measurements to section 7. In principle, I think (possibly different from Referee #2) that an extensive discussion about the interpretation of measurements should take place in a review. In particular, if defective interpretations are possible, this discussion is exactly where the authors can make an outstanding point by demonstrating the improvements obtained from their work.

6. Quasi global and high resolution observations with MetOp/IASI:

Please, make a statement about temporal coverage of IASI data, its availability and how to access IASI data. I do not agree with Referee #2, that Figs. 8 and 9 are redundant. It is of interest to see the seasonal change on the global map. But, I suggest to put Figs. 8 and 9 into one Figure with 4 panels. A discussion about the influence of small-scale variability (e.g., thermals) within a 12km large area is lacking.

7. Summary:

The authors summarize quiet well their progress in H2O-delta_D products. But, they should also point out the meaning for atmospheric research. What do we learn from the improvements in tracking water isotopologues? Maybe the (new) interpretations made in section 5 (criticized by Referee #2) are better discussed here in order to point out the gain for amtmospheric research. The authors also should describe the benefit

of the new quasi-global high-resolution product with more emphasis. "Conclusions" should be added to the sections' title.

Specific comments:

All contour plots: The Rayleigh-line should be plotted stronger (same thickness as black lines).

P7 L32: 5 0/00.

P7 L18 - L25: Please, give an estimate of the impact of atmospheric variability to the significance of the remote sensing products. This is a major issue, just mentioning here is not enough. The discussion should take place here and not in the appendix C. Details of flight data, measurement schedules and so on, of course, should be left there.

P8 L3: ...the procedure...

---

## Author Comment (AC1) · 26 Feb 2016

Dear referees,

Thank you very much for your collaboration and your comments on our manuscript. We have elaborated our replies and hope that they are satisfying for you. Please give us a feedback where you would like to have further explanations. There is still time until 14$^{th}$ March for a public discussion, if you like.

Best regards,
Matthias Schneider

REFEREE #1 COMMENT:
The manuscript AMT-2015-330 is a review of the development of the MUSICA framework for observing H2O-delta_D pairs and a report of its current state. The authors describe very detailed their solutions of the lacks in the calibration and validation of the retrievals, partially published already in previous articles. The manuscript is within the scope of the journal AMT. A review of the evolution in this intricate research field merits publication. But, as this being a review article, I expect the content being presented in a more general context, and, maybe with less details of the own work, that has been published before.

OUR REPLY:
Yes, we decided to make it a review article however, our aim is not a general review article of all the work that has been done in the context of tropospheric water vapour isotopologue observations. As stated already in the title, abstract, and introduction, it is a review/summary of the MUSICA activities (i.e. our own work during the last 5-10 years). In a general review we could not address all the outcomes of the MUSICA project with the necessary detail. However, we feel that exactly this is important, because with the MUSICA funding we had the unique opportunity to focus on the data consistency and quality assurance/documentation (less science but a lot of important technical outcomes, which are important for more robust scientific studies). For this review/summary it has been our intention to report the latest and final MUSICA developments, thereby complementing previous MUSICA publications. We tried to do this as clear as possible and our intention was to avoid unnecessary details. So we agree with the referee, that details should be avoided wherever one could refer to previous published work. However, occasionally we have a different opinion on where details are necessary and where not.

REFEREE #1 COMMENT:
1. Introduction:
A more general introduction into the research field of atmospheric water vapour isotopologues within the context of atmospheric dynamics should be added. The introduction currently reads more as an excuse for the authors work. A wider overview should introduce the reader into this topic and should set the authors work to less profile. This means also, the content of Appendix A should be shifted into the introduction, while Appendix A could be erased. There was previous work published (e.g. Worden et al., ACP 2011). This should be cited correctly. The authors generally should set their work more into the context. Less noisy advertising would be more satisfactory for a review.

OUR REPLY:

Yes, we agree: we should improve the scientific motivation of tropospheric water vapour isotopologues observations (we will try to improve the first two paragraphs in the Introduction section).

However, since we clearly state in the title, abstract, and introduction that we perform a review of the MUSICA project, we think that Appendix A should better remain an appendix. We think that for the Introduction section a brief review on the state-of-the-art of tropospheric water vapour isotopologue remote sensing observation is enough detail and this is given in paragraph 3 and 4. There we refer to the most important works: the first TES paper of Worden et. al., 2006; the Nature TES paper: Worden et al., 2007; the TES paper that describes the improved algorithm: Worden et al., 2012; the IASI retrievals made by the French/Belgium group: Lacour et al., 2012; the SCIAMACHY and GOSAT retrievals: Frankenberg et al., 2009; Frankenberg et al., 2013; Boesch et al., 2013; ground-based TCCON-like retrievals: Rokotyan et al., 2014; etc.) Furthermore, we discuss scientific studies using the different data products: e.g. Noone 2012; Berkelhammer et al., 2012; Risi et al., 2012a,b; Sutano et al. 2015; Field et al., 2015; etc. In case we miss something, we are happy to expand this list, please let us know.

In paragraph 5 we explain why it is so important to have a detailed look on the nature and quality of the water isotopologue remote sensing data, and we argue that relevant work is still not satisfactory. We will improve this paragraph and, as suggested by the referee, we will add an additional reference to the Worden at al. 2011 paper (this paper describes an indirect method of validating the TES data by means of point measurements at Mauna Loa). Furthermore, we will add references to the Herman et al., 2014 paper (aircraft isotopologue profiles from surface to about 4800 m a.s.l.) and to Lacour et al., 2015 (cross-comparison of TES and IASI).

In paragraph 6 we put MUSICA into the context and we will try to improve this paragraph.

When talking about "noisy advertising" we guess that the referee refers to the following sentence: "Actually we are only aware of one campaign (the summer 2013 MUSICA campaign, Dyroff et al., 2015), where such profiles are measured in coincidence with ground- and space-based observations and over the wide altitude range where the remote sensors are sensitive". We will explain better, why these profiles are indeed unique for calibrating the remote sensors. To our knowledge, these are the only water isotopologue in-situ profiles measured up to almost 7000 m a.s.l. and in coincidence to remote sensing observations (compared to 5000 m a.s.l. of other campaigns).

REFEREE #1 COMMENT:
2. Remote sensing of H2O and delta_D:
The authors describe their retrievals in great detail, which is acceptable here. They quantify their obtained errors, but, a general discussion about error sources is missing. This should necessarily be added.

OUR REPLY:
Error sources and methods for analytic error propagation have been presented and discussed in detail in Schneider et al., 2012 and Wiegele et al,. 2014. We thought it is sufficient to refer here to these previous publications. However, it is no problem to give here also some brief review on the leading error sources.

REFEREE #1 COMMENT:
3. Accurate remote sensing products:

This section presents the main results of the development of MUSICA remote sensing products and claims their high quality and accuracy. For evaluating the results, correlation coefficients and slope-values should be added to the tables 2 and 3. A discussion about the limits of the described products is largely missing. Appendix C points out that coincidence of the reference measurements was limited. But the impact of short term and small scale variability of tropospheric humidity is not discussed or even quantified. To my feeling a detailed discussion about this is necessary for any evaluation based on reference data with a temporal and spatial mismatch. Quantitative analyses of tropospheric variability of water vapour have recently been published in ACP (Steinke et al., 2015 and Vogelmann et al. 2015). These studies should be considered and cited.

It is stated, that one error source is the unknown humidity above the ceiling altitude of the aircraft. For me, this seems not to be that clear. Above 7km I would expect definitely less than 5% of the total water vapour column and consequently variations being even significantly smaller which is in a certain contrast to the named errors. However, I would expect larger uncertainties caused by short term and small scale variability.

OUR REPLY:
We would like to remark that Section 4 is equally important or even more important, since it validates the H2O-deltaD pairs. These pairs are the most useful parameter for tropospheric humidity research (not individual H2O and deltaD observations).

Yes, we will add correlation coefficients and slope values and discuss the limits of this "remote sensing data calibration". Actually, there are a lot of studies that investigate the magnitude of temporal and spatial mismatches. For instance the studies cited by the referee, but also our studies made for exaxtly the site (Tenerife), where the comparisons are made.

Our studies made for Tenerife:
In Wiegele et al. (2014) we estimated the impact of temporal and spatial mismatches on the comparisons of different datasets. The impact of temporal mismatch can be estimated by using the NDACC/FTIR data, which we often obtain continuously (one measurement every 10 minutes) during several hours. Within 2 hours we observed a scatter of typically 4% and 7.5‰ for middle tropospheric H2O and deltaD, respectively. In Schneider et al. 2010 (http://www.atmos-meas-tech.net/3/1785/2010/amt-3-1785-2010.html), we analysed FTIR H2O "profiles" measured with an even higher temporal resolution (one measurement each 2 minutes). Within 2 hours we again find a scatter for middle tropospheric H2O of about 5% (for 3 hours it is about 10%, see Schneider et al. 2010, Fig. 12). However, close to the surface H2O varies by 10% already within 1 hour. The good agreement within 2 hours was already reported in the Schneider et al. 2009 study (http://www.atmos-meas-tech.net/3/323/2010/amt-3-323-2010.html, e.g. Fig. 4 therein).

The studies cited by the referee made for other sites:
The good correlation of humidity measurements made within 3 hours has also been reported by Steinke et al. 2015 (for instance Fig. 7 therein). Vogelmann et al. 2015 adds DIAL measurements to such studies, so that mismatches in the profile can be better estimated. In their Figure 6 they report middle tropospheric (4-6 km) H2O variabilities within 2 hours of about 20% and within 1 hour of about 15%. This variability increases strongly in the upper troposphere where they found about 60% within 2 hours. When interpreting this study, we have to take into account that the DIAL detects vertically high resolution profiles, whereas the FTIR and IASI remote sensing data represent deep layers (where variability cancels partly out).

Studies for estimating spatial mismatch:
The impact of spatial variability can be estimated by space-based observations or by models. Steinke et al. 2015 used ICON simulations and estimated the scatter for humidity encountered at a distance of 10 km to be about 4% (right column in their Fig. 4 using an IWV of 12 kg/m^2 from their Fig. 2). In Wiegele et al. 2014 we estimated a variability of about 19% and 17‰ for

middle tropospheric H2O and deltaD measured within an 110 km x 110 km area around Tenerife, whereby at least half of this variability is due to the random errors in the IASI data.
Summary:
In agreement to a variety of different studies we can assume an uncertainty of about 5% and 7.5‰ for H2O and deltaD, respectively, due to temporal mismatches for comparisons made in the middle troposphere within 2 hours. For comparisons in the lower troposphere the uncertainty is larger. We discuss this limitation in detail in Appendix C and document it in Fig. 12. A reasonable estimate of the uncertainty coming from spatial mismatches can be estimated to be 10% and 10‰ for H2O and deltaD, respectively. We will improve these discussions in Appendix C by adding additional references and in Section 3 briefly summarize the discussion in context of the limitations of our calibration study.
Unknown humidity above the ceiling altitude is important. Please be aware that the retrievals are performed on the logarithmic scale in the volume mixing ratio domain. The averaging kernels for 5 km have significant values between 7 and 10 km (i.e., altitudes for which we do not know the humidity and the isotopologue ratios do affect the retrievals at 5 km). Concrete values are given in Appendix C2.

REFEREE #1 COMMENT:
4. Validation of H2O and delta_D pairs:
I agree with Referee #2. This section is a somewhat extensive and could be more a summary of the previous work (Wiegele 2014, Schneider 2015).

OUR REPLY:
Yes, we agree. The description is too detailed here, especially since these pathways have been already discussed in Schneider et al. 2015 and González et al. 2015. We will remove most of the details, keep Table 4 as a brief summary and refer for more details to Schneider et al. 2015 and González et al. 2015.
Figures 4-6 are a central outcome of the project MUSICA and they must be kept in the paper. These figures importantly complement what has been shown in previous MUSICA publications. Furthermore, the different pathways we identified for the subtropical Atlantic offer a very good opportunity for validating space-based remote sensing products of H2O-deltaD pairs. It is very important to show this in a clear way and to allow other groups to perform a similar validation exercises for other space-based H2O-deltaD data products. Please see also the attachment at the end of this reply where we again would like to demonstrate the complementarity of the Figs. 4-6 to figures shown in previous publications.

REFEREE #1 COMMENT:
5. Consistent long-term observation with NDACC/FTIR:
The contour plots of Fig. 7 show, that there is a seasonality in H20-delta_D-pairs, but I would expect at least something like a long time series for different locations. Such a plot should have the time-scale on the x-axis. It might be better, to move the interpretation of measurements to section 7. In principle, I think (possibly different from Referee #2) that an extensive discussion about the interpretation of measurements should take place in a review. In particular, if defective interpretations are possible, this discussion is exactly where the authors can make an outstanding point by demonstrating the improvements obtained from their work.

OUR REPLY:
In Schneider at al., 2012 we show such time series (Figure 12 therein), although only for 1996 to 2011. Such plot can be useful to present the isotopologue dataset and we can refer to the

time series plot in that paper instead of showing a new time series plot (repeating this plot would in our opinion be redundant in our opinion). For scientific applications H2O-deltaD distribution plots are more interesting and our idea is to show the usefulness of the long-term data for studying the seasonal behavior of the H2O-deltaD distribution. And we think that Fig. 7 should be kept in this section. We will explain better the scientific usefulness of investigating seasonal behavior of H2O-deltaD: gaining insight in water sources, pathways for different seasons might help for a better understanding of the developments of droughts (e.g. the current drought in East Africa: is it linked to anomalies in H2O-deltaD? And can this help to better understand the development/reasons for the drought?).

REFEREE #1 COMMENT:
6. Quasi global and high resolution observations with MetOp/IASI:
Please, make a statement about temporal coverage of IASI data, its availability and how to access IASI data. I do not agree with Referee #2, that Figs. 8 and 9 are redundant. It is of interest to see the seasonal change on the global map. But, I suggest to put Figs. 8 and 9 into one Figure with 4 panels. A discussion about the influence of small-scale variability (e.g., thermals) within a 12km large area is lacking.

OUR REPLY:
Yes, the referee is right. It is important to clarify here that MUSICA MetOp/IASI data are currently only available for a very limited number of observations. We have the product generated on global scale and for IASI A and B, but only for 6 days in February 2014 and for 12 days in August 2014. In addition, we have made retrievals for 2007-2013 for a 2 degree x 2 degree area around Tenerife (Canary Island) and Kiruna (Northern Sweden) as well as for Karlsruhe, Germany (for Karlsruhe only for 2010 to 2013). The data can be requested by contacting the authors. Depending on future funding we will intensify the retrievals and provide the data via a database. Our clear intention is to make a significant amount of retrievals and in this context to collaborate closely with EUMETSAT.
Yes, there is no problem in combining Figs 8+9 to a single figure.
In Wiegele et al. 2014 we made a detailed error estimation. Among others we estimated the impact of a partial cloudy pixel on the retrieval. Is this what the referee means by "small scale variability (e.g. thermals)"?

REFEREE #1 COMMENT:
7. Summary:
The authors summarize quiet well their progress in H2O-delta_D products. But, they should also point out the meaning for atmospheric research. What do we learn from the improvements in tracking water isotopologues? Maybe the (new) interpretations made in section 5 (criticized by Referee #2) are better discussed here in order to point out the gain for atmospheric research. The authors also should describe the benefit of the new quasi-global high-resolution product with more emphasis. "Conclusions" should be added to the sections' title.

OUR REPLY:
Yes, we agree. We will improve the "Summary and Conclusion" section and add a paragraph where we describe the scientific progress that might be achieved by using tropospheric water vapour isotopologue observations.
We strongly prefer to discuss the scientific usefulness of the NDACC/FTIR and MetOp/IASI H2O-deltaD observations individually in their respective sections (Section 5 and 6).

NDACC/FTIR is better suited for climatological considerations and IASI/MetOp for investigating global patterns.

REFEREE #1 SECIFIC COMMENTS:
- All contour plots: The Rayleigh-line should be plotted stronger (same thickness as black lines).
- P7 L32: 5 0/00.

OUR REPLY:
Yes, thanks! We agree.

REFEREE #1 SECIFIC COMMENTS:
- P7 L18 - L25: Please, give an estimate of the impact of atmospheric variability to the significance of the remote sensing products. This is a major issue, just mentioning here is not enough. The discussion should take place here and not in the appendix C. Details of flight data, measurement schedules and so on, of course, should be left there.

OUR REPLY:
Yes, we agree that this can be improved. As stated in one of the replies to the General Comments we will improve/expand the respective discussion in Appendix C and give an improved summary of this discussion in Sect. 3.

REFEREE #1 SECIFIC COMMENTS:
- P8 L3: ...the procedure...

OUR REPLY:
Yes, thanks! We agree.

The paper "A framework for accurate, long-term, global and high resolution observations of tropospheric H2O-_D pairs - a MUSICA review" presents the results of the MUSICA project which aims at providing coherent _D-H2O observations from different platforms. While no new scientific progress is reported, the publication of this paper as a review paper makes sense for potential users of _D because of the complex nature of such remote sensed products. Moreover a summary paper on the different available products would nicely help the potential users. I would however strongly recommend to the authors to make changes and cuts in order to make it more like a review paper. Some sections provide redundant information which was already provided in your previous papers. It would be nice to illustrate the long term aspect of your FTIR data by showing long term time series. It would also be useful for potential users to document the capabilities of all the FTIR stations in term of vertical sensitivity (averaging kernels and DOFS, I did not find this information in your previous papers).

OUR REPLY:
Yes, we agree: this new paper should not provide redundant information already given in previous papers! Referee #1 had a similar comment and again, we would like to remark that exactly this has been our intention. However, it seems that occasionally there are different opinions about what is a detail that can be looked up in previous publications and what is complementary to previous publications and thereby important for this review paper.

REFEREE #2 COMMENT:
Section 2 - Retrieval description:
You spend some time describing the retrieval setup, it would be also useful to report the noise levels you used for the FTIR and IASI retrievals. Also you state that you fit the temperature from IASI spectra (section 2.3.2), is it the same procedure than in Wiegele et al., 2014 (a priori is the EUMETSAT L2 with a strong constraint)? In that case it would be interesting to report it as well because it means you are sensitive to EUMETSAT temperature errors.

OUR REPLY:
We tried to keep the description of the retrieval as short as possible and focus on the modification made with respect to previous work. However, it is no problem to make a more detailed description and repeat here the important retrieval details as mentioned by the referee: (1) our cost functions work with the noise as seen in the residual (measured – simulated spectrum). (2) Our IASI temperature retrieval is strongly constrained towards the EUMETSAT L2 temperature (this has been not changed with respect to what has been reported in the Schneider and Hase 2011 and the Wiegele et al. 2014 papers). As a consequence, our retrieval product will be sensitive to the EUMETSAT IASI L2 temperature error.

REFEREE #2 COMMENT:
Section 3: Accurate remote sensing products
- The use of accurate here suggests that there is no error in the retrievals, the authors however nicely describe a systematic bias between certain ranges. The description of a most likely range of bias is very useful for the users but it also indicates that the retrievals are not exact like the title suggests. I recommend to modify the title to a less overselling one, such as "absolute characterization of remote sensing products".
- Could you provide the numbers in the description of the correlation (correlation coefficient and associated p-values). Section 3.1 L18 and Section 3.2 L6

- The comparison of retrieval products with non-smoothed in situ data is also of interest for the users as it allows to realize how much the smoothing affects the data. Could you add a comparison between non-smoothed data (like you do in Figure 5) and retrieved ones or say a word about how the smoothing affects the in situ data.

OUR REPLY:
- Yes, we agree and will rename the title of the section as suggested by the referee: "Absolute characterisation of remote sensing products"
- Yes, no problem, we will provide the correlation coefficients and slope values
- Yes, we agree. It is important to understand how the smoothing affects the actual atmospheric state. However, this has been shown and discussed in great detail in the Schneider et al. 2015 paper (see Figs. 6 and 8 therein, compare left and central panels). We will add one sentence where we refer to these figures.

REFEREE #2 COMMENT:
Section 4: Validation of H2O-deltaD pairs
It is true validating the added value of deltaD retrieval is very important but it has already been made in Schneider et al., 2015 and Wiegele et al., 2014. What is the interest to present here the moisture pathways and the extreme approaches again? The novelty compared to these previous papers is not clear. Moreover the message of these two approaches is the same. I recommend presenting only one or the other. Is there a reason why you would expect differences between IASI-A and IASI-B? Are the radiances measurements not calibrated and compared? Could you develop why you need to verify.

OUR REPLY:
We agree that the description of the different moisture pathways and the extreme approach is maybe too detailed here, because we could refer to previous papers. The different moisture pathways are well described in González et al. 2015 (to a less extent in Schneider et al., 2015) and the extreme approach is well explained in Wiegele et al. 2015. So we will shorten this description. However, we also want to insist that the figures in Section 4 are a central outcome of MSICA and clearly complement the plots as shown in previous MUSICA publications. Please have a look on the Attachment at the end of these replies, where we explain in detail the complementarity.
The moisture pathway approach is made for the subtropics. This is a very clear study and compares the remote sensing H2O-deltaD distribution (affected by the averaging kernels, i.e. a smoothed representation of the atmospheric state) with the in-situ H2O-deltaD distribution (absolute calibrated and not affected by averaging kernels, i.e. a direct representation of the atmospheric state). The extreme approach compares remote sensing H2O-deltaD distributions. So it presents no comparison to absolute references and both remote sesning data do only represent a smoothed version of the actual atmospheric state. However, the extreme approach is complementary because it is made for different locations (subtropics, mid-latitudes and subpolar region), whereas the moisture pathway approach is only possible for the subtropics (only there we have middle tropospheric in-situ and remote sensing observations).
We think that the comparison of H2O-deltaD products obtained from IASI A and B is important. The referee is of course right that the radiances of both instruments are well calibrated and we use the same retrieval. So theoretically differences should not be expected. However, we think it is important to see these expectations actually confirmed. We compare the observations made by the two IASI instruments within a 0.25 x 0.25 degree area. For such coincidences both instruments have often very different viewing geometries (different swath angles) and they detect different airmasses (slight spatial mismatch). So our IASI A/B comparison gives further

confidence in the retrieval process (consistent results for different swath angles), again confirms that a spatial mismatch within 50 km (pixel center within a 0.25 x 0.25 area) and temporal mismatch within 1 hour does not significantly affect the comparisons (important for Section 3, see also comment of referee #1) and might also address the concerns of "small scale variability (e.g. thermals)" mentioned by referee #1.

REFEREE #2 COMMENT:
Section 5: Consistent long-term observation with NDACC/FTIR
- It would be useful in the perspective of a review paper on the capabilities of the MUSICA network to present the vertical sensitivities of the different FTIR sites. What vertical sensitivity can we expect from the new FTIR? Maybe within the Table 5, you could add a column with the altitude range of sensitivity and the dofs.
- An example (or several) of long term times series would nicely illustrate the title of the section. Is there any inter annual variability, do you observe trends? Such questions would greatly help to assess the added value of such long term timeseries.
- The lower panel of Figure 7 shouldn't appear in this section, it is more related to section 4 but the message was already clear from section 4.1.

OUR REPLY:
- Yes, we agree. We will add in Table 5 the typical DOFS values. And we will also add a sentence referring to Table 6 and Figs. 8 and 9 (left panels) of Schneider et al. 2012, where these sensitivities are presented and discussed in more detail.
- For a scientific exploitation of the lower/middle tropospheric isotopologue data analyses of H2O-deltaD distributions are best suited. In this section we show an example of how continuous long-term data might help to understand climatological issues (like the seasonal behavior). Time series of H2O and deltaD have already been shown in Schneider et al. 2012 (Fig. 12 therein) and we feel that it is redundant to show it again in the review paper.
- We think that it is very important to make the community aware of the risks of data misuse. The remote sensing isotopologue products are complex and H2O-deltaD plots might be misinterpreted if the data are not post processed. It is true that this has already been discussed in Section 4, but here we show the impact of these data misuse discussing a concrete example. From the comment of referee #1 we have the feeling that he/she shares our opinion in this regard.

REFEREE #2 COMMENT:
Section 6: Quasi global and high resolution observations with IASI
- The processing of all IASI data is very challenging and requires the development of dedicated solution, are you able to process the IASI data in near real time? It is also an information that data users are interested in and that would fit within this section.
- Figure 8 and 9 are redundant, one is enough to illustrate your point.
- The type 1 versus type 2 discussion would better fit in section 4.

OUR REPLY:
- Yes, we agree. Here we must clarify that we are currently not able to do an operational processing of the IASI data. We have plans to process larger amounts of data, however at the moment there are only a few days of globally processed data available and long-term data processing has only been made for limited areas around three ground-based FTIR sites.
- Similar to referee #1, we think that it is important to document the typical global coverage for winter and summer, so we think that Figs. 8 and 9 are not redundant.

- In Section 6 we give an example for a scientific study with IASI H2O-deltaD data pairs. Using an example study that shows how data misuse (type 1 versus type 2) can lead to important misinterpretations is very important. Our impression is that referee #1 has a similar opinion. We think that we should have this discussion where we discuss the potential of IASI by means of the example study (diurnal signal over the Sahara).

REFEREE #2 COMMENT:
Appendix D
This appendix could be summarized in the article in one sentence without associated figure. I understand that the coverage is slightly better for type I product but I believe one number would be enough to illustrate that.

OUR REPLY:
Ok, we will remove Appendix D and just mention the better coverage for the Type 1 product in the paper in one/two sentences.

REFEREE #2 SPECIFIC COMMENT:
- "Exemplary": often used throughout the manuscript, if possible, alternate
- P2, L14: deltaD x 1000

OUR REPLY:
Ok, we agree, the manuscript will be improved accordingly.

REFEREE #2 SPECIFIC COMMENT:
- P2 L31: "for investigating the dynamics of the MJO (..)", add Tuinenburg et al., 2015 as a reference concerning the study of MJO with IASI.

OUR REPLY:
Yes, we will add it. The Tuinenburg et al. 2015 study is a further example that a better discussion on the risk of remote sensing H2O-deltaD data misuse is needed. In Tuinenberg et al. 2015 IASI type 2 products are used for analyzing the relation between H2O-deltaD. It is found that the slopes are different to the slopes found by the analyses of Berkelhammer et al 2012. Since Berkelhammer et al. 2012 used TES H2O-deltaD products with the charactistics of a type 1 product, it might well be that using type 2 versus type 1 explains the differences in the H2O-deltaD slopes obtained in the two studies.
So again we would like to insist on the importance of the Figs. 4, 7 and 10 of our paper (these figures give examples of such risk).

REFEREE #2 SPECIFIC COMMENT:
- P2, L33: "However, there is only one study so far where attempts have been made for empirically validating these H2O-_D pairs (Schneider et al.,2015)" – Herman et al., 2014 and Worden et al., 2011 are previous attempts to validate TES deltaD retrievals, please consider them in this sentence.
- P2, L35: "Further and more detailed validation efforts for the H2O-_D pairs are urgently needed"- Urgently sounds a bit over dramatic regarding the numerous publications that have been made since the TES data (2006) for example.

- P5, L24: "The Type 2 product offers consistent H2O-_D pairs, which are sensitive to the lower and the middle troposphere, whereby it is possible to reasonably separate both altitude regions (degree of freedom for signal, DOFS, is typically about 1.8)." – Is it the case for all FTIR sites? What are the dofs for the different sites? This information would be useful in the table 5.

OUR REPLY:
Ok, we agree, the manuscript will be improved accordingly.

REFEREE #2 SPECIFIC COMMENT:
- P7, L2: "We would like to remind that there is no similar study where the accuracy of simultaneous tropospheric H2O and _D remote sensing observations is documented by such direct comparison to coincident reference profiles". – What about Herman et al., 2014?

OUR REPLY:
We would like to make a difference to the Herman et al. 2014 study, since they use profiles that only reach altitudes below 5000 m a.s.l. This strongly compromises the absolute calibration of the remote sensing data, which are strongly affected by the atmosphere above 5000 m a.s.l. This is still a problem with our aircraft data, even though they reach much higher altitudes (almost 7000 m a.s.l.). We will better explain this here and also refer to the Appendix C2.

REFEREE #2 SPECIFIC COMMENT:
- P7, L18: Please specify the coefficient correlation values
- P8, L2: The figure mentioned here shows spatial colocation of the different measurements and does not give any information on the bias. The figure is not needed in this context, referencing to Schneider et al (2015) is sufficient.
- P12,L18-22: "the MUSICA NDACC/FTIR isotopologue data are representative for different altitudes and for larger scale processes (the data represent vertical layers averaged over 2-5 km (see typical averaging kernels in Fig. 3 of Schneider et al., 2015) "- Again, are these "typical" averaging kernels representative of all the FTIR stations? A large scale process better defines a process that occurs at a wide horizontal scale than a vertical one. I suggest using "deeper layers".

OUR REPLY:
Ok, we agree, the manuscript will be shortened/improved accordingly.

REFEREE #2 SPECIFIC COMMENT:
- P13, L4:10: " After the rain season (OND, blue) the vapour is most depleted in HDO. Then evaporation of precipitated surface water might be more important as free tropospheric moisture source than during the other seasons. Between January and March (JAM, green) the vapour concentrations remain similar as during the previous months, but the air becomes more enriched in HDO, which might indicate increased importance of mixing with humid boundary layer air and reduced importance of condensation processes. For April to June (AMJ, purple) air gets more humid, but _D remains almost constant, indicating to further moistening due to even stronger mixing with humid boundary layer air." - The interpretation of _D is not straitghtforward and several combinations of different processes can explain a same H2O- _D composition. I would keep this part very brief to avoid erroneous interpretations. For example you suggest that the surface evaporation in OND will give water vapor a more depleted signature. However,

evaporation from surface tends to enrich water vapor as there is no fractionation during evapotranspiration. Or in MAM, different airmass trajectories from different sources with same mixing historic could explain the observed signatures.

OUR REPLY:
Ok, the referee is right. The interpretation of H2O-deltaD plots is very difficult and without models or back-trajectory analyses it is very difficult to draw robust conclusions. We tried to give a reasonable explication of the observed seasonality, but this is actually speculative and we will remove it or at least clearly mention the speculative character of it.

REFEREE #2 SPECIFIC COMMENT:
- P13, L15: "The IASI ground pixel at nadir has a diameter of only 12 km" – 12 km is not that small, GOSAT as TES ground pixels are smaller, moreover there is less nadir measurement than not nadir measurement because of the large IASI swath, remove only.
- P13,L23: "Please recall that one data point represents a ground pixel of 12 km diameter (at nadir)" – This sentence is not necessary as it was said a few lines before.
- P17,L2: IASI and TES retrievals have been cross-validated in Lacour et al., 2015 you should add this reference here. Also add the reference in the Table 6.

OUR REPLY:
Ok, we agree, the manuscript will be shortened/improved accordingly.

**Attachment: Explanation of the progress in the validation of H2O-δD pairs (Section 4 of the new paper)**

In the following we show the progress in the H2O-δD pair validation with respect to the validation as shown in Wiegele et al. (2014) and Schneider et al. (2015).

**Figure 4 and respective discussions:**

H2O-δD analysis as in Schneider et al. (2015), but on logarithmic scale:

[Figure]

The Figure above presents the data as in Schneider et al. (2015). The only difference with Schneider et al. (2015) is the usage of a logarithmic instead of a linear H2O-scale.

In the following we show Figures that describe the advances (1-4, step by step) made for the new paper.

(1) H2O-δD analysis with calibrated H2O-δD data from v2015 and only for sufficient sensitivity at 5 km:

[Figure]

It is obvious that the calibration and filtering significantly changes the distribution of H2O-δD data pairs. We cannot expect that the H2O-δD pair validation as shown in Schneider et al. (2015) for the previous retrievals is also valid for the final retrievals. We need to be very careful!

(2) Improved H2O-δD analysis by using density distributions:

[Figure]

For the new paper we refined the approach for validating the $H_2O$-δD pairs. We calculate density plots on the $\ln[H_2O]$-δD surface. This gives a more detailed and realistic picture. For instance, the density areas as plotted in the Figure above are made with the same data points as depicted in the previous Figure. While in the density plot we see that there is a rather low probability of data points being situated close to the Rayleigh line, the simple data point plot suggests even significant probability of data points being below the Rayleigh line. In consequence, the introduction of density plots is a significant advance.

(3) Refined classification of atmospheric situations:

A further refinement has been made in the context of the classification of the moisture transport pathways. The improved classification is possible by using backward trajectories in addition to dust observations as proxy for the moisture pathways. The improved classification method has been presented in Gonzaléz el al. (2015) for the in-situ data. In this new paper we apply the new method for the first time to remote sensing data (this is a new development and it has not been used for the study as shown in Schneider et al., 2015).

[Figure]

By the improved classification, the value of the remote sensing $H_2O$-$\delta D$ is much better documented! In the previous plot the density curves for SAL and no SAL move largely in parallel to a Rayleigh line and one could argue that $\delta D$ adds actually nothing new, because we simply observe lower $\delta D$ for lower $H_2O$. Now with the improved classification we can clearly prove that this is not the case. We have now three different groups and there are situations where the $H_2O$-$\delta D$ distribution changes parallel and perpendicular to a Rayleigh line. The $H_2O$-$\delta D$ distributions for these different groups are similar in the remote sensing and the in-situ data. This is of course an extremely important advance if compared to the Schneider et al. (2015) paper! With this improved validation it is now very clear that the $H_2O$-$\delta D$ distributions obtained from the remote sensing data sets can be used for diagnosing different moisture pathways. There has been no other similar study so far! Actually, we think that this validation should be required for any $H_2O$-$\delta D$ remotes sensing pair before being used for moisture pathway research.

(4) Be careful with misinterpretations (possible problems when using Type 1 data):

[Figure]

For the Figure above the same retrieval results as in the previous Figure are used. The only difference is the aposteriori processing, which has not been performed for the here plotted data. We observe that the density plots are very different. This has never been shown before. In Schneider et al. (2012), Wiegele et al. (2014), and Schneider et al. (2015) we theoretically discuss the importance of the aposteriori processing. In this new paper it is the first time that we show with real data that for a correct interpretation of the $H_2O$-$\delta D$ plots the usage of aposteriori processed data is highly recommendable. This is a very important demonstration in particular since there are several studies that draw conclusions from the $H_2O$-$\delta D$ distributions as observed in remote sensing data. It is very important to make the community aware of the risk for misleading conclusions.

Figure 5 and respective discussion:

It is important to repeat the comparison of the IASI and FTIR $H_2O$-$\delta D$ pairs for the final data versions. We cannot draw conclusions from the respective Izana plot as shown in Wiegele et al. (2014), because the new retrieval version changes the distribution of $H_2O$-$\delta D$ pairs (see discussion above concerning the different H2O-$\delta D$ distributions for the previous and v2015 data versions).

The data shown in Wiegele et al. (2014) are for different altitudes (5 km for Izana and 2.4 km for Karlsruhe and Kiruna) and for each site we use a different apriori. Now we compare for all three sites the same altitudes and use the same apriori. From the new plot we can clearly conclude that the two datasets correctly capture the differences between the three sites in a similar manner. From Tenerife via Karlsruhe to Kiruna the relative difference to the unique apriori as well as the ln[$H_2O$]- $\delta D$ slopes change in a similar manner. This is an important validation result, which is in no way observable in a plot as shown in Wiegele et al. (2014).

Figure 6, global scale comparison of H2O-$\delta D$ pairs between IASI-A and IASI-B:

There is no other study that performs such comprehensive comparison of H2O-$\delta D$ pairs measured by two different space-based instruments! This Figure clearly gives further confidence in the retrieval process (consistent results for different swath angles) and again confirms that a spatial mismatch within 50 km (pixel center within 0.25 x 0.25 area) and temporal mismatch within 1 hour does not significantly affect the comparisons.